# Response to interferons and antibacterial innate immunity in the absence of tyrosine-phosphorylated STAT1

Andrea Majoros[1], Ekaterini Platanitis[1,†], Daniel Szappanos[1,†], HyeonJoo Cheon[2], Claus Vogl[3], Priyank Shukla[3], George R Stark[2], Veronika Sexl[4], Robert Schreiber[5], Christian Schindler[6], Mathias Müller[3] & Thomas Decker[1,*]

## Abstract

Signal transducer and activator of transcription 1 (STAT1) plays a pivotal role in the innate immune system by directing the transcriptional response to interferons (IFNs). STAT1 is activated by Janus kinase (JAK)-mediated phosphorylation of Y701. To determine whether STAT1 contributes to cellular responses without this phosphorylation event, we generated mice with Y701 mutated to a phenylalanine (Stat1$^{Y701F}$). We show that heterozygous mice do not exhibit a dominant-negative phenotype. Homozygous Stat1$^{Y701F}$ mice show a profound reduction in Stat1 expression, highlighting an important role for basal IFN-dependent signaling. The rapid transcriptional response to type I IFN (IFN-I) and type II IFN (IFNγ) was absent in Stat1$^{Y701F}$ cells. Intriguingly, STAT1Y701F suppresses the delayed expression of IFN-I-stimulated genes (ISG) observed in Stat1$^{-/-}$ cells, mediated by the STAT2/IRF9 complex. Thus, Stat1$^{Y701F}$ macrophages are more susceptible to *Legionella pneumophila* infection than Stat1$^{-/-}$ macrophages. *Listeria monocytogenes* grew less robustly in Stat1$^{Y701F}$ macrophages and mice compared to Stat1$^{-/-}$ counterparts, but STAT1Y701F is not sufficient to rescue the animals. Our studies are consistent with a potential contribution of Y701-unphosphorylated STAT1 to innate antibacterial immunity.

**Keywords** innate immunity; interferon; pathogen; phosphorylation; STAT1
**Subject Categories** Immunology; Microbiology, Virology & Host Pathogen Interaction

## Introduction

The Jak-Stat signaling pathway regulates cellular responses to cytokines. The prototypic signal transducer and activator of transcription, STAT1, plays a crucial role in host defense by mediating the effects of interferons (IFNs). In the canonical signaling pathway, all types of IFNs produce transcriptionally active STAT1 through Janus kinase (JAK)-mediated phosphorylation at Y701. The type II interferon (IFNγ) receptor complex phosphorylates STAT1 exclusively, thus producing homodimers of the transcription factor. These translocate to the cell nucleus and stimulate gene expression through binding to gamma interferon-activated sequences (GAS) within the IFN response regions of its target genes. By contrast, stimulation with type I or type III interferons (IFN-I and IFN-III, respectively) produces phosphorylated STAT1–STAT2 heterodimers. These heterodimers associate with interferon regulatory factor 9 (IRF9) to form the IFN-stimulated gene factor 3 (ISGF3). After translocation to the nucleus, ISGF3 binds to interferon-stimulated response elements (ISREs) to stimulate gene expression [1,2].

In addition to canonical, tyrosine-phosphorylated ISGF3 and STAT1 dimers, STATs 1 and 2 form non-canonical complexes. Transcriptional responses by STAT1/IRF9 complexes in complete absence of STAT2 or in the absence of its phosphorylation, and by STAT2 complexes lacking STAT1, are documented in the literature [3–8]. Non-canonical complexes may contain transcriptionally active STATs without a phosphotyrosine (U-STATs) [8–10]. In *Drosophila*, U-STATs associate with and maintain the stability of heterochromatin. Whether similar examples of this chromatin/STAT link exist also in *Dictyostelium* and *C. elegans*, organisms that appear to lack JAKs, but still have STATs, is currently not known [11–13]. Recent experimental evidence supports the concept that U-STAT1 prolongs the expression of a subset of IFN-induced genes,

1   Max F. Perutz Laboratories, University of Vienna, Vienna, Austria
2   Department of Molecular Genetics and Proteomics Core, Lerner Research Institute, Cleveland Clinic, Cleveland, OH, USA
3   Institute of Animal Breeding and Genetics, University of Veterinary Medicine Vienna, Vienna, Austria
4   Department for Biomedical Sciences, Institute of Pharmacology and Toxicology, University of Veterinary Medicine Vienna, Vienna, Austria
5   Department of Pathology and Immunology, Washington University School of Medicine, St Louis, MO, USA
6   Departments of Microbiology & Immunology and Medicine, Columbia University, New York, NY, USA
    *Corresponding author. Tel: +43 1 4277 54605; Fax: +43 1 4277 9546; E-mail: thomas.decker@univie.ac.at
    †These authors contributed equally to this study

many of which are involved in immune regulation [14]. It has been hypothesized that this occurs due to the accumulation of newly synthesized STAT1 through a positive feedback loop after interferon stimulation. Furthermore, it has been proposed that prolonged exposure of cells to IFNβ induces the expression of non-phosphorylated STAT2 (U-STAT2) and IRF9 which combine with U-STAT1 and form the unphosphorylated ISGF3 (U-ISGF3) [9]. U-ISGF3 in turn maintains the expression of a subset of the initially induced ISGs, leading to extended resistance to virus infection and DNA damage.

In the present manuscript, we investigate the role of U-STAT1 signaling in mice expressing a Stat1$^{Y701F}$ mutant. We report an important contribution of STAT1 tyrosine phosphorylation to a tonic signal increasing Stat1 expression. Extensive analysis of STAT activation and type I IFN-induced genes (ISGs) expression patterns demonstrate clear differences between macrophages lacking STAT1 and those expressing STAT1Y701F. These are reflected by an altered ability to limit the growth of the intracellular bacterial pathogen *Le. pneumophila*. In spite of reduced STAT1Y701F protein amounts, our work suggests a minor but significant contribution of STAT1Y701F also to the innate response against another bacterial pathogen, *L. monocytogenes*. In summary, our results provide evidence for biological activity of STAT1 in the absence of its tyrosine phosphorylation.

# Results

## Jak-Stat signaling in Stat1$^{Y701F}$ mice

To investigate STAT1 activity in the absence of its tyrosine phosphorylation, we generated Stat1$^{Y701F}$ knock-in mice by targeting the gene with a construct encoding phenylalanine in position 701 (see Materials and Methods). We first examined heterozygous Stat1$^{Y701F/+}$ mice. Importance of this genetic configuration stems from human patients, where Stat1$^{Y701C}$ heterozygosity was classified as an autosomal dominant cause of Mendelian susceptibility to mycobacterial disease (MSMD) [15,16]. We observed reduced Y701 phosphorylation of STAT1 expressed from the WT allele in macrophages treated with either IFNβ or IFNγ (Fig 1A). The kinetics of STAT1 phosphorylation were comparable to those of wild-type cells (Fig 1A). Reduced STAT1 phosphorylation on Y701 corresponded to decreased protein amounts. STAT2 phosphorylation by IFNβ and total STAT2 amounts were normal. In line with diminished STAT1 tyrosine phosphorylation, maximal expression of interferon-stimulated genes (ISGs) was lower after stimulation with either IFNβ (Fig 1B) or IFNγ (Fig 1C). The data suggest that Stat1$^{Y701F}$ heterozygosity causes reduced STAT1 expression and a correspondingly lower activation by the interferon receptors.

When analyzing macrophages from mice carrying homozygous Stat1$^{Y701F}$ mutation for STAT1 protein, we noted strongly diminished amounts of total STAT1 (Fig 2A, far right panel) and treatment with either IFNβ or IFNγ showed the expected loss of tyrosine phosphorylation (Fig 2B and E). Strongly reduced amounts of STAT1 were also observed in livers, spleens, and other organs of mutant animals (Fig 2A). Reportedly, small amounts of IFN-I are secreted and accumulate in tissue milieu even in the absence of infection. Constitutive IFN-I secretion was proposed to generate a tonic signal and cause a priming effect keeping cells in

a state of alertness to other cytokines and contributing to immune homeostasis, maintenance of bone density, and antiviral and antitumor immunity [17]. Our data provide the first *in vivo* evidence that a tonic signal, most likely from the IFN-I receptor, increases Stat1 expression through a mechanism involving its phosphorylation on Y701. Intriguingly, the levels of constitutive STAT1 phosphorylation are below the radar of the tools used for detection in most cell types, yet they are sufficient to exert a strong biological impact.

In accordance with expectations, the early transcriptional response to interferons after 4 h was absent in Stat1$^{-/-}$ as well as Stat1$^{Y701F}$ cells (Fig 2C and D). Previous studies in Stat1$^{-/-}$ cells demonstrated the occurrence of STAT1-independent, STAT2-dependent gene expression at a delayed stage of the transcriptional response to IFN-I [4,6,18–20]. Consistent with our previous report [3], stimulation with IFNβ caused STAT1-independent ISG expression starting around 8–12 h after treatment which required the presence of both STAT2 and IRF9 (Fig 2D). Intriguingly, the presence of STAT1Y701F partially repressed STAT2/IRF9-dependent, STAT1-independent genes at late stages of the IFNβ response. This relationship between genotype and expression profile was noted for many other investigated ISGs (Fig EV1B). The presence of STAT1Y701F inhibited the late expression of these genes through the STAT2/IRF9-dependent pathway.

After treatment with IFNγ, both Stat1$^{Y701F}$ and Stat1$^{-/-}$ macrophages failed to induce a transcriptional response of typical STAT1 target genes, such as Irf1, CIIta, Stat1, and Stat2 at any time after treatment (Figs 2C and EV1A). Consistent with their regulation by STAT1 homodimers, lack of STAT2 or IRF9 had no or little impact on their expression (data not shown). These findings suggest that inhibition of late-stage gene expression by STAT1Y701F is selective for the response to IFN-I.

## Identical STAT2 tyrosine phosphorylation profile in Stat1$^{-/-}$ and Stat1$^{Y701F}$ macrophages

In order to further characterize the mechanism by which STAT1Y701F represses STAT1-independent, STAT2/IRF9-dependent gene expression during the late IFN-I response, we considered a dominant-negative effect of the mutant at the level of JAK-mediated tyrosine phosphorylation. Therefore, tyrosine phosphorylation of STATs was profiled by Western blot. In accordance with [3], STAT1 showed both delayed and prolonged phosphorylation in Stat2$^{-/-}$ and Irf9$^{-/-}$ macrophages stimulated with IFNβ compared to WT (Fig 2E). 24 h of stimulation with IFNβ leads to upregulation of total STAT1 levels equally well in WT, Stat2$^{-/-}$, and Irf9$^{-/-}$, while it remained absent in Stat1$^{Y701F}$ mutant macrophages (Figs 2E and EV1B). We also observed delayed, but prolonged phosphorylation of STAT2 in Stat1$^{Y701F}$, Stat1$^{-/-}$, and Irf9$^{-/-}$ macrophages. Similar to STAT1, the total amounts of STAT2 were upregulated after 24 h of stimulation with IFNβ in all genotypes, but never as well as in WT, with Stat1$^{Y701F}$ mutant having the weakest induction (Figs 2E and EV1B). The data suggest that the presence or absence of ISGF3 subunits exerts strong influence on both the tyrosine phosphorylation and upregulation of STATs 1 and 2.

Apart from STAT1, STAT3 and STAT5 are known to contribute to tissue-specific interferon signaling [21]. To examine possible effects of STAT1Y701F on other STATs, we profiled the tyrosine

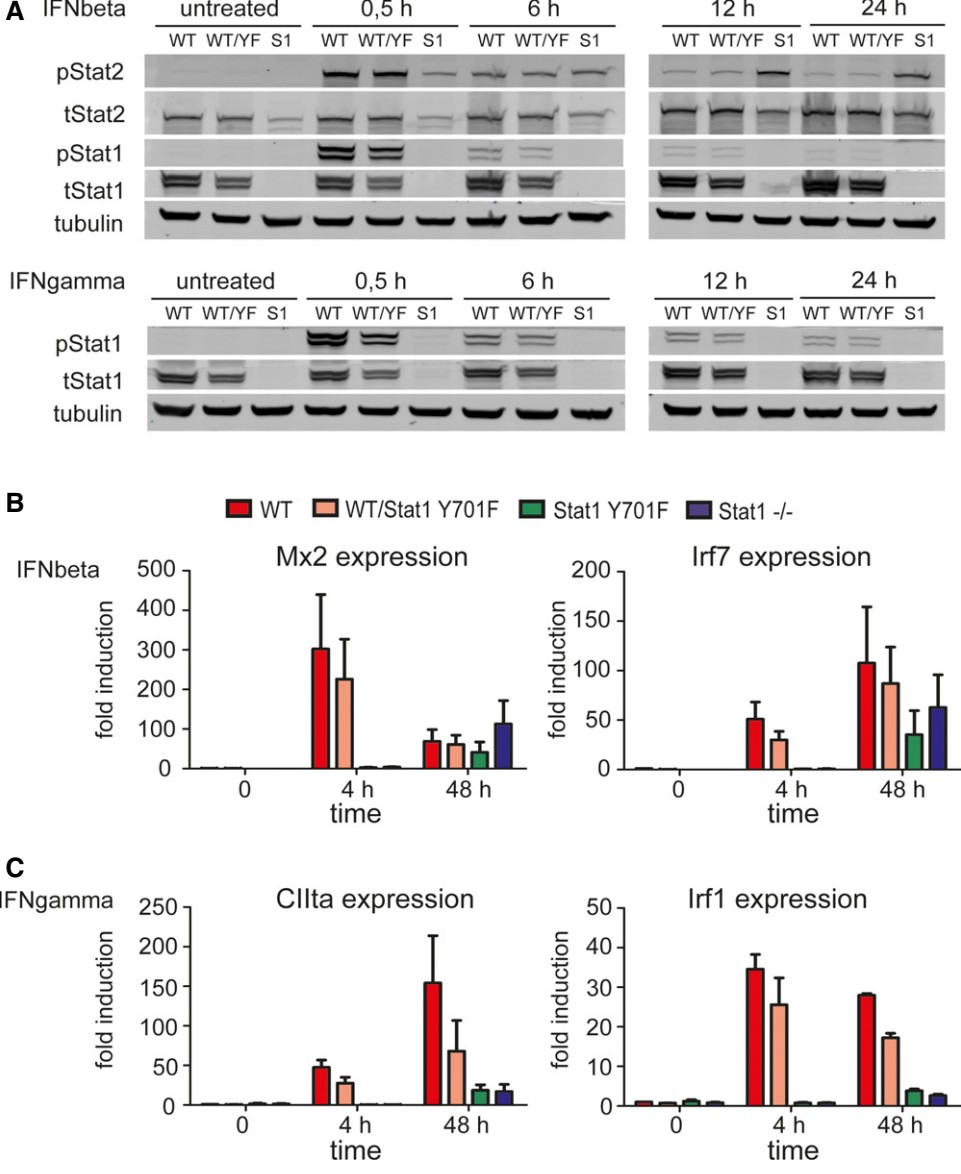

**Figure 1. Interferon signaling in mice bearing heterozygous Stat1[Y701F] mutation resembles that of human cells with heterozygous Stat1[Y701C] mutation.**

A   Western blot analysis of STAT expression and phosphorylation. Bone marrow-derived macrophages (BMDMs) of wild-type (WT), Stat1[Y701F/+] (WT/YF), or Stat1[−/−] (S1) mice were treated with 250 IU/ml of IFNβ or 5 ng/ml of IFNγ for 0.5, 6, 12, and 24 h. Whole-cell extracts were collected and tested in Western blot for levels of phosphorylation of STAT1 (Y701) and STAT2 (Y689) and total level of STAT1 and STAT2. The blots are representative of more than three independent experiments.

B   Effect of Stat1[Y701F] heterozygosity on the expression of type I IFN-induced genes (ISGs). BMDMs of wild-type (WT), Stat1[Y701F/+] (WT/YF), or Stat1[−/−] (S1) mice were treated with 250 IU/ml of IFNβ for 4 and 48 h. Gene expression was measured by qPCR and normalized to Gapdh and to the expression levels in untreated wild-type cells. The bars represent mean values with the standard deviations (SD) of three independent experiments.

C   Effect of Stat1[Y701F] heterozygosity on the expression of IFNβ-induced genes. BMDMs of wild-type (WT), Stat1[Y701F/+] (WT/YF), or Stat1[−/−] (S1) mice were treated with 5 ng/ml of IFNγ for 4 and 48 h. Gene expression was measured by qPCR and normalized to Gapdh and to the expression levels in untreated wild-type cells. The bars represent mean values with the standard deviations (SD) of three independent experiments.

phosphorylation of STAT3 and STAT5 by Western blot. Compared to their WT counterparts, cells expressing STAT1Y701F displayed an unimpaired ability to phosphorylate STAT3 and STAT5 in response to IFNβ (Fig EV2). In accordance with this, amounts of STAT3 and STAT5 proteins were unchanged (Fig EV2). These data confirm the importance of STAT2 phosphorylation and signaling in STAT1-independent, STAT2/IRF9-dependent gene expression, but did not explain the inhibition of the STAT2/IRF9 pathway by STAT1Y701F.

## Cytoplasmic STAT1Y701F inhibits STAT1-independent STAT2 signaling

According to the Jak-Stat paradigm, STATs need to be phosphorylated on tyrosine to perform nuclear functions. This is also true for the U-Stat pathway proposed by Stark and coworkers because U-Stat function follows an early, tyrosine-dependent IFN response [14]. However, data with several STATs suggest a cytoplasmic or

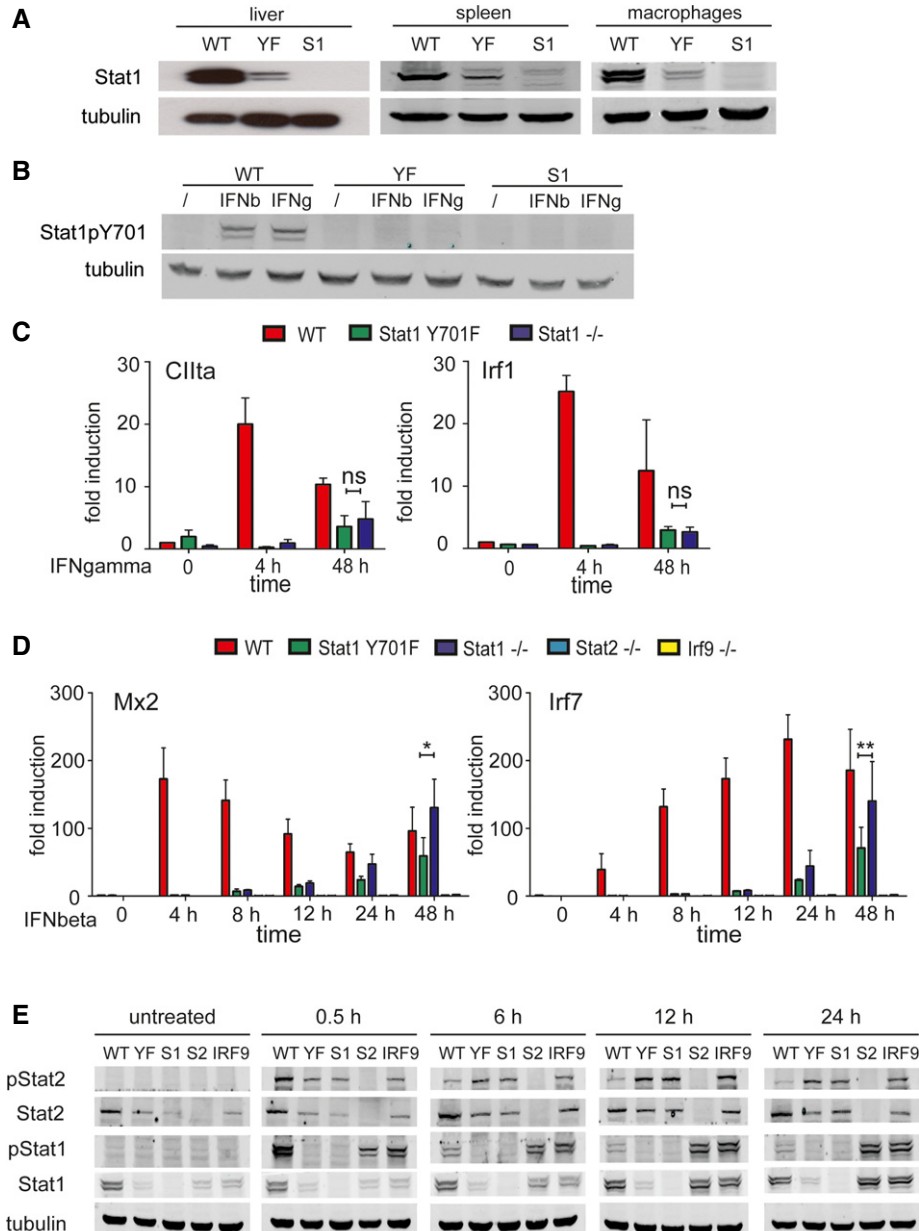

**Figure 2.  STAT1 expression and interferon signaling in Stat1$^{Y701F}$ mice.**

A   Effect of Stat1$^{Y701F}$ homozygosity on STAT1 expression in mouse cells and organs. Spleens, livers, or bone marrow-derived macrophages (BMDMs) were isolated from wild-type (WT), Stat1$^{Y701F}$ (YF), and Stat1$^{-/-}$ (S1) mice. Whole-cell extracts were collected and tested for levels of total STAT1 in Western blot. The blots are representative of more than three independent experiments.

B   Effect of Stat1$^{Y701F}$ homozygosity on STAT1 phosphorylation at Y701. BMDMs were isolated from wild-type (WT), Stat1$^{Y701F}$ (YF), and Stat1$^{-/-}$ (S1) mice and stimulated for 30 min with 250 IU/ml of IFNβ or 5 ng/ml of IFNγ. Whole-cell extracts were collected and tested for levels of STAT1 phosphorylation on Y701 in Western blot. The blots are representative of more than three independent experiments.

C   Effect of Stat1$^{Y701F}$ homozygosity on the expression of IFNβ-induced genes. BMDMs of wild-type (WT), Stat1$^{Y701F}$ (YF), and Stat1$^{-/-}$ (S1) mice were treated with 5 ng/ml of IFNγ for 4 or 48 h. Gene expression was measured by qPCR and normalized to Gapdh and to the expression levels in untreated wild-type cells. Bars represent a mean value of three independent experiments. Error bars represent standard error of the mean (SEM); asterisks denote the level of statistical significance (ns, $P > 0.05$); and the *P*-values were calculated using paired ratio *t*-test.

D   Effect of Stat1$^{Y701F}$ homozygosity on the expression of type I IFN-induced genes (ISGs). BMDMs were isolated from wild-type (WT), Stat1$^{Y701F}$, Stat1$^{-/-}$, Stat2$^{-/-}$, and Irf9$^{-/-}$ mice treated with 250 IU/ml of IFNβ for 4, 8, 12, 24, or 48 h. Gene expression was measured by qPCR and normalized to Gapdh and to the expression levels in untreated wild-type cells. Bars represent a mean value of three independent experiments. Error bars represent standard error of the mean (SEM); asterisks denote the level of statistical significance (*$P \leq 0.05$; **$P \leq 0.01$); and the *P*-values were calculated using paired ratio *t*-test.

E   STAT1 and STAT2 phosphorylation in Stat1$^{-/-}$, Stat1$^{Y701F}$, Stat2$^{-/-}$, and Irf9$^{-/-}$ macrophages. BMDMs were isolated from wild-type (WT), Stat1$^{Y701F}$ (YF), Stat1$^{-/-}$ (S1), Stat2$^{-/-}$ (S2), and Irf9$^{-/-}$ (IRF9) mice and treated with 250 IU/ml of IFNβ for 30 min or 6, 12, or 24 h. Whole-cell extracts were collected and tested in Western blot for levels of phosphorylation of STAT1 on Y701 and of STAT2 on Y689. The same cell extracts were tested for total levels of STAT1 and STAT2. The blots are representative of more than three independent experiments.

    

organelle-based function independently of tyrosine phosphorylation [22–28].

To examine the nuclear effect of STAT1Y701F, next-generation sequencing of chromatin immunoprecipitations with antibodies to STAT1 (ChIP-Seq) was carried out. Both untreated and IFNβ-stimulated macrophages of WT and Stat1$^{Y701F}$ genotypes were analyzed. The data provide no hint of nuclear presence of STAT1Y701F before or after IFNβ treatment. By contrast, binding of WT STAT1 was readily observed after IFNβ treatment (Fig EV3 shows Mx2, Irf7, Stat1, and Stat2 genes as representative examples).

Consistent with the suppression of gene expression stimulated by STAT2/IRF9 complexes, site-directed ChIP showed decreased STAT2 binding to both Mx2 and Irf7 ISRE sequences in Stat1$^{Y701F}$ cells 24 h after IFNβ treatment (Fig 3A). In WT macrophages, STAT1 occupied the Mx2 ISRE site both 2 and 24 h after stimulation with IFNβ and binding required STAT2 (Fig 3B). ChIP-reChIP experiments confirmed the simultaneous presence of STAT1 and STAT2 24 h after IFNβ stimulation on at least a subfraction of ISREs, suggesting that in WT cells late-stage induction involves both STAT1 and STAT2 (Fig 3C). Further evidence for a suppression of STAT2/IRF9-mediated gene expression was obtained with the experiment shown in Fig 3D. Transfection of STAT2 into STAT1-deficient fibroblasts stimulated late IFNβ-induced gene expression. Expression of STAT1Y701F along with STAT2 suppressed gene expression when compared to the amount obtained with STAT2 overexpression alone. By contrast, WT STAT1 enhanced gene expression when transfected together with STAT2.

In the absence of evidence for nuclear STAT1Y701F activity, we performed immunofluorescence-based experiments to test whether the lack of individual ISGF3 subunits influences nuclear translocation of STAT2. Staining for STAT2 showed a comparable translocation of STAT2 to the nucleus 30 min after IFNβ stimulation of Stat1$^{-/-}$ and Irf9$^{-/-}$ macrophages (Fig 4A). Twenty-four hours after IFNβ stimulation, STAT2 was largely present in the nucleus of Stat1$^{-/-}$ and Irf9$^{-/-}$ macrophages. Importantly, the presence of STAT2 in the nucleus of Stat1$^{Y701F}$ macrophages was reduced in comparison with Stat1$^{-/-}$ (Fig 4B).

Together, the data strongly suggest that STAT1Y701F leads to inhibition of STAT1-independent, STAT2/IRF9-dependent late ISG expression through interaction with STAT2 in the cytoplasm, preventing it from nuclear translocation and DNA binding.

## Stat1$^{Y701F}$ and Stat1$^{-/-}$ macrophages and mice differ in their response to infection with *Legionella pneumophila* and *Listeria monocytogenes*

To examine whether suppression of late IFN-I signaling by STAT1Y701F alters cell-autonomous antibacterial immunity provided by macrophages, we infected the cells with two intracellular pathogens, *Le. pneumophila* and *L. monocytogenes*. While IFN-I was shown to limit intracellular growth of *Le. pneumophila* with a clear impact of the STAT1-independent delayed pathway [29–31], this activity of IFN-I is not seen upon infection with *L. monocytogenes* [32].

Infection of untreated macrophages with *Le. pneumophila* showed no difference between WT, Stat1$^{-/-}$, or Stat1$^{Y701F}$ genotypes (Fig 5A). As previously reported [3,30], *Le. pneumophila* growth was reduced after IFN-I treatment in both WT and Stat1$^{-/-}$ macrophages (Fig 5B). Consistent with the suppressive activity of STAT1Y701F on the STAT1-independent pathway, Stat1$^{Y701F}$ macrophages showed less ability than Stat1$^{-/-}$ to inhibit *Le. pneumophila* growth (Fig 5B).

During infection of WT macrophages with *L. monocytogenes*, IFN-I is produced and causes phosphorylation of STAT1 on Y701 (Fig 6A), but IFN-I does not affect growth in WT or Stat1-deficient macrophages [32]. We tested whether Stat1$^{Y701F}$ mutation and STAT1-deficiency have a different impact on the transcriptional response to *L. monocytogenes* by performing gene microarrays using cDNA from infected cells. Comparing Stat1$^{Y701F}$ macrophages with WT, we found differentially expressed probes at an FDR of 0.1 at all observed time points during infection (Table 1, first row). In striking contrast to the comparison with the WT, no differentially regulated genes at an FDR of 0.1 were found at any time point when we compared Stat1$^{Y701F}$ and Stat1$^{-/-}$ (Table 1, second row). Among the genes higher expressed in WT versus Stat1$^{-/-}$ as well as Stat1$^{Y701F}$ macrophages was the Ifnb gene (Fig 6B). Thus, the strongly reduced production of type I IFN is likely to obscure potential effects IFN-I might exert on the innate response of Stat1$^{Y701F}$ cells and mice to *L. monocytogenes*. It may also preclude significant alterations of gene expression when compared to STAT1-deficiency.

To investigate potential effects of unphosphorylated STAT1, we infected macrophages *in vitro* and determined intracellular *L. monocytogenes* growth. Strikingly, untreated macrophages with the Stat1$^{Y701F}$ genotype inhibited *L. monocytogenes* replication better than BMDM with complete STAT1-deficiency (Fig 7A), suggesting a role for U-STAT1.

STAT1 is essential for innate resistance of mice against *L. monocytogenes* infection [33,34]. Prompted by the result in isolated macrophages, we determined the impact of STAT1Y701F in murine *L. monocytogenes* infection. Due to the strong decrease in STAT1 amounts caused by Y701F mutation with respect to WT, the most important readout of these experiments is a potential gain of function in comparison with Stat1$^{-/-}$ mice.

Stat1$^{Y701F}$ mice infected with *L. monocytogenes* were highly susceptible to infection when compared to WT, but showed reduced bacterial loads in lungs, brain, liver, and spleen 48 h and 72 h after infection compared to the Stat1$^{-/-}$ mice (Fig 7C). The difference between the Stat1$^{-/-}$ and Stat1$^{Y701F}$ genotypes was smaller at 72 h after infection and disappeared at the terminal stage of infection shortly before death (between 72 and 144 h p.i.). Consistently, Stat1$^{Y701F}$ and Stat1$^{-/-}$ mice died at equal rates (Fig 7B). We conclude that compared to STAT1-deficiency, the presence of STAT1Y701F delays *L. monocytogenes* replication, however, without the efficacy that would be needed for an increase in survival. The delay in bacterial spread and the concomitant generation of inflammatory infiltrates was further supported by immunohistochemistry. Stat1$^{Y701F}$ mice livers contained fewer *L. monocytogenes* (Fig 8A) and immune cell infiltrates were smaller both in numbers and size compared to the Stat1$^{-/-}$ mice livers (Fig 8A and B). As previously described [35], these infiltrates consisted mostly of neutrophils (Fig 8A), while there were no clearly discernable differences in macrophage distribution in livers of all three genotypes (Fig EV4). Furthermore, measurements of alanine aminotransferase (ALT) levels in serum of infected mice showed a decrease of this liver damage parameter in Stat1$^{Y701F}$ mice compared to Stat1$^{-/-}$ (Fig 8C). The data demonstrate a subtle, yet

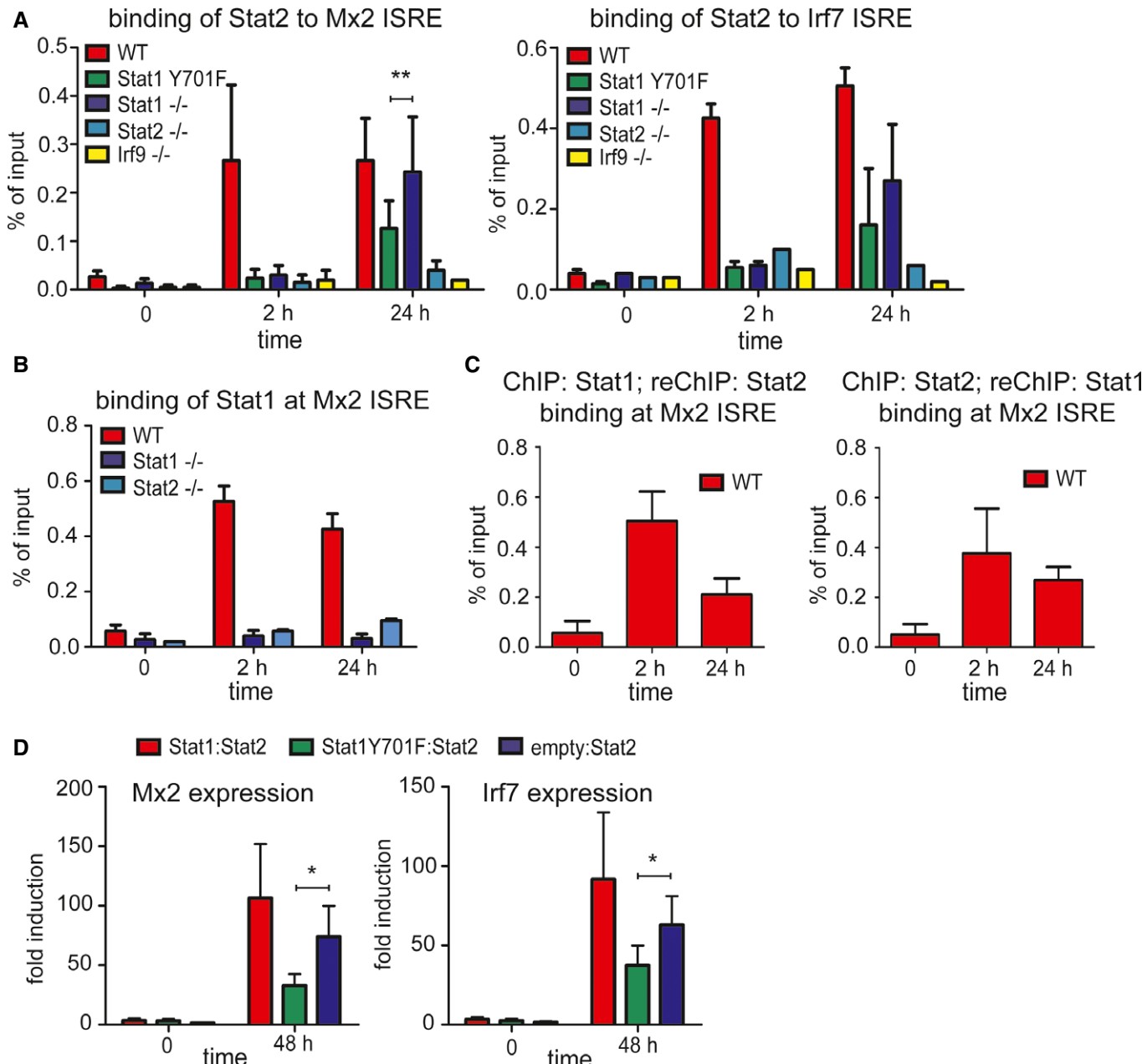

**Figure 3.  Presence of the STAT1Y701F mutant reduces IFNβ-stimulated binding of STAT2 to nuclear ISRE sequences.**

A   IFNβ-stimulated binding of STAT2 to ISRE sequences of Mx2 and Irf7 promoters. Bone marrow-derived macrophages (BMDMs) of wild-type (WT), Stat1$^{Y701F}$ (YF), Stat1$^{-/-}$ (S1), Stat2$^{-/-}$ (S2), and Irf9$^{-/-}$ (IRF9) mice were treated with 250 IU/ml of IFNβ for 2 or 24 h. Cells were cross-linked, sonicated, and immunoprecipitated with STAT2-specific antibody. The amount of precipitated DNA was measured by qPCR. Bars represent a mean value of three independent experiments. Error bars represent standard error of the mean (SEM); asterisks denote the level of statistical significance (**P ≤ 0.01); and the P-values were calculated using paired ratio t-test.

B   Impact of STAT2 deficiency on IFNβ-stimulated STAT1 association with the Mx2 ISRE. BMDMs of wild-type (WT), Stat1$^{-/-}$ (S1), and Stat2$^{-/-}$ (S2) mice were treated with 250 IU/ml of IFNβ for 2 or 24 h. Cells were cross-linked, sonicated, and immunoprecipitated with STAT1-specific antibody. The amount of precipitated DNA was measured by qPCR. Bars represent mean values of three independent experiments; error bars represent standard error of the mean (SEM).

C   Simultaneous association of STAT1 and STAT2 with the Mx2 ISRE analyzed by ChIP-reChIP. BMDMs of wild-type (WT) mice were treated with 250 IU/ml of IFNβ for 2 or 24 h. Cells were cross-linked, sonicated, and immunoprecipitated with either STAT1-specific antibody and re-immunoprecipitated with STAT2-specific antibody or vice versa. The amount of precipitated DNA was measured by qPCR. Bars represent mean values of three independent experiments; error bars represent standard deviation (SD).

D   Impact of STAT1Y701F on delayed, STAT2-mediated expression of IFN-induced genes. Stat1$^{-/-}$ fibroblasts were transfected with plasmids driving expression of the indicated proteins. Twenty-four hours after transfection, 250 IU/ml of IFNβ was added to the transfected cells and ISG expression was determined by qPCR after 48 h of cytokine treatment. Gene expression was measured by qPCR and normalized to Gapdh. Bars represent a mean value of three independent experiments. Error bars represent standard error of the mean (SEM) and asterisks denote the level of statistical significance (*P ≤ 0.05); the P-values were calculated using paired ratio t-test.

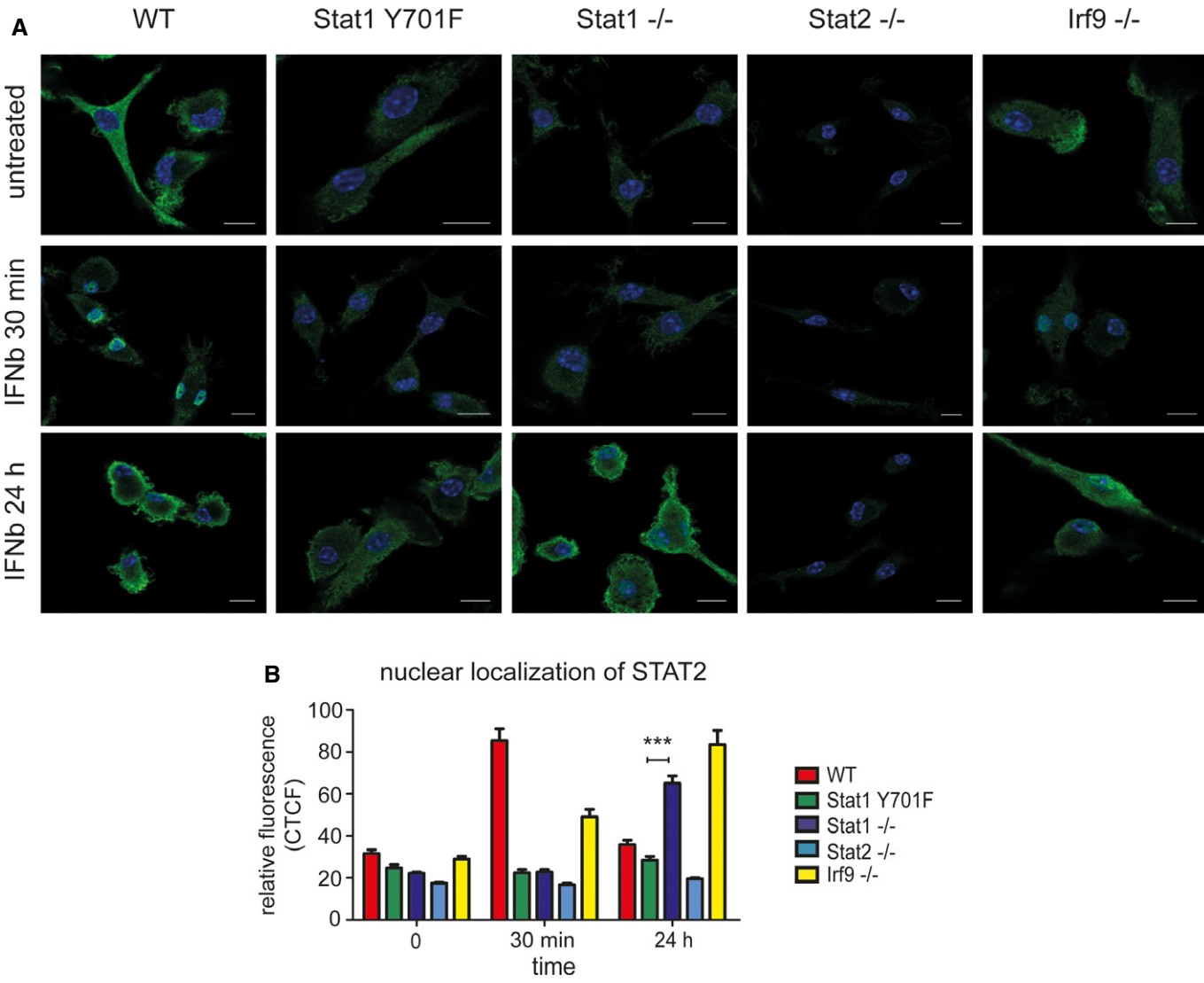

**Figure 4.   STAT1Y701F mutant reduces IFNβ-stimulated nuclear translocation of STAT2.**

A   Analysis of STAT2 nuclear translocation by immunofluorescence. Bone marrow-derived macrophages (BMDMs) of wild-type (WT), Stat1$^{Y701F}$ (YF), Stat1$^{-/-}$ (S1), Stat2$^{-/-}$ (S2), and Irf9$^{-/-}$ (IRF9) mice were seeded on cover slips and stimulated with 250 IU/ml of IFNβ for 30 min or 24 h. The cells were fixed and stained for STAT2-specific antibody followed by Alexa Fluor® 488 conjugated secondary antibody (green). Nuclei were stained with DAPI (blue). Studies are representative of more than three independent experiments. The scale bars represent 10 μm.

B   Quantitative evaluation of STAT2 nuclear translocation. The intensity of STAT2-dependent immunofluorescence over DNA staining (DAPI) was quantified using ImageJ software in 20 cells from two independent experiments. Bars represent a mean with standard deviation (SD) and asterisks denote the level of statistical significance (***$P \leq 0.001$); $P$-value was calculated using unpaired $t$-test.

clearly discernible activity of U-STAT1 in the innate response against *L. monocytogenes*.

# Discussion

STAT1 makes an essential contribution to innate immunity against viral and intramacrophagic bacterial disease. The objective of our study was to reveal any contribution of non-tyrosine-phosphorylated STAT1 to innate immunity. Examining cells and organs of mice expressing a Stat1$^{Y701F}$ mutant, the first observation of note was the strong dependence of STAT1 expression on its tyrosine

phosphorylation through tonic signaling. None of the tested stimuli, including *L. monocytogenes* infection, caused upregulation of the Stat1 gene in the absence of its tyrosine-phosphorylated product. Therefore, effects of the U-Stat pathway as defined by Stark and colleagues which rely on an increase in STAT abundance could not be examined. In addition, any results obtained for the immune response of Stat1$^{Y701F}$ cells or mice could not be compared to WT counterparts because a distinction between effects of tyrosine phosphorylation and effects of STAT1 abundance was not possible. To overcome this problem, we recorded gain or loss of function with respect to Stat1$^{-/-}$ cells and mice. With this approach, we were able to derive important insight into the cross-regulation of Stat1 and

    

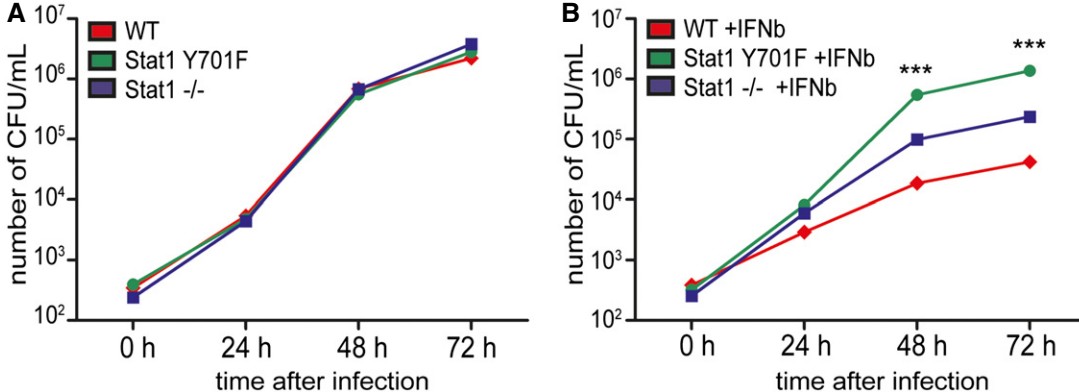

**Figure 5. STAT1Y701F mutant counteracts the inhibition of *Legionella pneumophila* replication by delayed, STAT2/IRF9-dependent IFN signaling.**

A *Legionella pneumophila* growth in unstimulated macrophages. Bone marrow-derived macrophages (BMDMs) of wild-type (WT), Stat1$^{Y701F}$, and Stat1$^{-/-}$ mice were seeded in 24-well plates and infected with *Le. pneumophila* (JR32 Fla−, MOI 0.25). The numbers of colony-forming units (CFUs) were determined 24, 48, and 72 h after infection on charcoal yeast extract plates (CYE). The 0 time point was collected 1.5 h after the infection.

B *Legionella pneumophila* growth in IFNβ-treated macrophages. Bone marrow-derived macrophages (BMDMs) of wild-type (WT), Stat1$^{Y701F}$, and Stat1$^{-/-}$ mice were seeded in 24-well plates, treated with 500 U/ml of IFNβ, and then infected with *Le. pneumophila* (JR32 Fla−, MOI 0.25). The numbers of colony-forming units (CFU) were determined 24, 48, and 72 h after infection on charcoal yeast extract plates (CYE). The 0 time point was collected 1.5 h after the infection.

Data information: The results in (A) and (B) represent six biological repeats and the data are presented as mean values. Asterisks denote statistically significant differences between CFU numbers from Stat1$^{Y701F}$ and Stat1$^{-/-}$ cells (***$P ≤ 0.001$); *P*-values were calculated using unpaired *t*-test.

Stat2 genes and the mechanism of Stat signaling by the IFN receptors. We also demonstrated inhibition of delayed, STAT1-independent type IFN signaling by the STAT1Y701F mutant, owing to inhibited nuclear translocation of STAT2. Reduced inhibition of intracellular *Le. pneumophila* growth in Stat1$^{Y701F}$ compared to Stat1$^{-/-}$ macrophages supports our interpretation of this activity. Other than *Le. pneumophila*, STAT1-independent type I IFN signaling is thought to partially rescue cells and organisms from infection with viruses causing STAT1 inhibition or degradation. Finally, we noted that the presence of STAT1Y701F inhibited *L. monocytogenes* replication, although not rescuing mice from lethal infection.

Mice expressing one Stat1$^{Y701F}$ allele in addition to a WT Stat1 allele showed reduced STAT1 expression, in accordance with phosphotyrosine-dependent autoregulation of the gene. Decreased tyrosine phosphorylation of WT STAT1 upon IFN treatment may thus reflect decreased protein amounts rather than a dominant-negative effect of STAT1Y701F. Several mutations of human Stat1 have been reported to impair the phosphorylation of STAT1 on Y701 [36–38]. However, only one clinical case of two-generation kindred has been described to have a heterozygous Y701C Stat1 mutation [16]. In this report, the patient's blood leukocytes showed normal STAT1 levels, but reduction in STAT1 phosphorylation and binding to DNA after both type I and type II interferon treatment. Therefore, Stat1$^{Y701C}$ mutation was concluded to be autosomal dominant. We cannot entirely rule out autosomal dominance of Stat1$^{Y701F}$ mutation in mice, but the reduced STAT1 expression appears to manifest a cell type- or species-specific difference.

In line with expectations, the early transcriptional response to both tested IFN types was completely lost in Stat1$^{Y701F}$ mice. Tyrosine phosphorylation of STATs 2, 3, and 5 did not differ from that observed in Stat1$^{-/-}$ cells. This shows the lack of dominant-negative activity of the Y701F mutant with regard to STAT activation by the IFN receptor complex. STAT2 levels were slightly elevated in

resting Stat1$^{Y701F}$ compared to Stat1$^{-/-}$ cells. However, upon IFNβ treatment STAT2 protein increased to higher levels in Stat1$^{-/-}$ cells, but much less so in Stat1$^{Y701F}$ cells or Irf9$^{-/-}$ cells. In agreement with data in Fig EV1, this shows that the Stat2 gene is a target of STAT1-independent STAT2/IRF9 signaling which, like many other genes examined in our study, is partially suppressed by STAT1Y701F. Regarding the mechanism of STAT1Y701F-mediated ISG suppression, our ChIP-Seq data support the conclusion that it is not through activity in the cell nucleus. While we cannot exclude insufficient sensitivity of this technology to detect very small STAT1Y701F amounts, a clear effect on STAT2 nuclear translocation suggests a cytoplasm-based mode of action. The finding that the small amounts of STAT1Y701F produce this significant effect is explained on the one hand by reduced nuclear presence of hemiphosphorylated STAT1Y701F/STAT2 dimers and by the resulting defect in STAT2 upregulation. STAT1Y701F/STAT2 dimers might either have reduced ability to enter the nucleus or to persist in this cell compartment through stable association with chromatin [39,40]. This would be expected to reduce the amount of gene-associated STAT2/IRF9 complexes, an interpretation compatible with both the analysis of STAT2 subcellular distribution (Fig 4A) and the site-directed ChIP demonstrating reduced STAT2 binding to ISRE-containing ISG promoters in Stat1$^{Y701F}$ cells (Fig 3A).

Unlike Stat2, the Stat1 gene increased profoundly after IFNβ treatment not only in WT, but also in all investigated genotypes except Stat1$^{Y701F}$. This result further emphasizes the importance of Y701 phosphorylation for Stat1 gene expression. Enhanced expression in Stat2$^{-/-}$ and Irf9$^{-/-}$ cells shows that IFNβ regulation of the Stat1 gene does not involve the ISGF3 complex. Surprisingly and in contrast, STAT1 maintenance in resting cells appears to require ISGF3 as a profound drop occurs in Stat2$^{-/-}$ as well as Irf9$^{-/-}$ macrophages.

Innate immunity to *L. monocytogenes* was strongly reduced in Stat1$^{-/-}$ as well as Stat1$^{Y701F}$ mice, in accordance with its

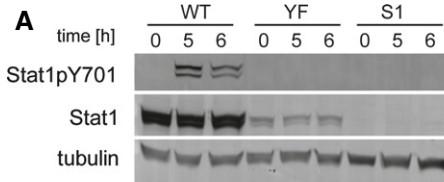

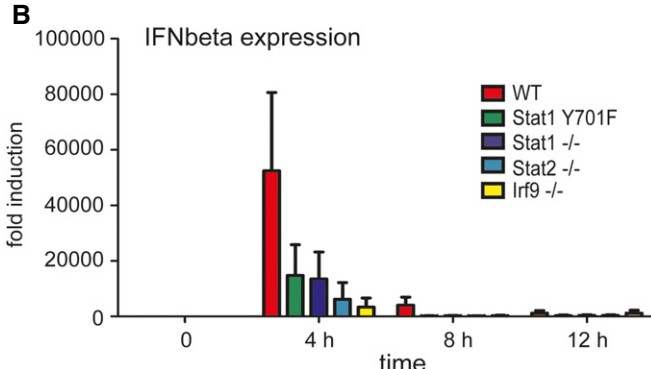

**Figure 6.**    **Interferon signaling in *Listeria monocytogenes*-infected Stat1[Y701F] bone marrow-derived macrophages.**

A    Western blot analysis of STAT1 tyrosine phosphorylation. Bone marrow-derived macrophages (BMDMs) of wild-type (WT), Stat1[Y701F] (YF), and Stat1[−/−] (S1) mice were infected with *L. monocytogenes* (LO28, MOI 10) for 5 or 6 h. Whole-cell extracts were collected and tested in Western blot for levels of total STAT1 and phosphorylation of STAT1 on tyrosine 701 (Y701). The blots are representative of more than three independent experiments.

B    Impact of Stat1[Y701F] mutation, or of deletion of ISGF3 subunits on the expression of the Ifnb gene. BMDMs of wild-type (WT), Stat1[Y701F], Stat1[−/−], Stat2[−/−], and Irf9[−/−] mice were infected with *L. monocytogenes* (LO28, MOI 10) for 4, 8, 12, 24, or 48 h. Levels of *Ifnb* gene expression were determined by qPCR. Bars represent mean values of three independent experiments. Error bars represent standard error of the mean (SEM).

**Table 1.    Transcriptome changes in macrophages of WT, Stat1[Y701F], and Stat1[−/−] genotypes after infection with *Listeria monocytogenes*.**

|  | Untreated | 6 h | 12 h | 24 h |
|---|---|---|---|---|
| WT | 6 (0.08%) | 1047 (13.68%) | 570 (7.46%) | 274 (3.58%) |
| Stat1[−/−] | 0 | 0 | 0 | 0 |

Numbers indicate differentially expressed genes between WT and Stat1[Y701F] macrophages (first row) and between Stat1[Y701F] and Stat1[−/−] macrophages (second row). Numbers in brackets indicate percent changes with respect to all genes analyzed. Microarray data are deposited at ArrayExpress under accession number E-MTAB-3598 (https://www.ebi.ac.uk/arrayexpress/experiments/E-MTAB-3598/).

dependence on IFNγ and the lack of any apparent transcriptional activity of STAT1Y701F on IFNγ-inducible genes (Figs 7A and EV3) [41,42]. In fact, STAT1Y701F was not able to rescue from lethal infection irrespective of the inoculum size (Fig 7B). In spite of this, replication of the bacteria in infected Stat1[Y701F] cells and organs was delayed, as were the examined hallmarks of liver inflammation and damage. At present, we cannot pinpoint the mechanism behind this. Speculatively, the altered response to IFN-I might be a contributing factor. Although Listeria-infected Stat1[−/−] and Stat1[Y701F] cells produce less IFN-I (Fig 6B), levels of cytokine are expected to

stimulate cells, albeit to a lesser degree. In the absence of IFN-I signaling, Listeria-infected mice show decreased bacterial burden and increased survival. This correlates with reduced apoptosis of splenic lymphocytes [43–46]. The impact of the late phase of the IFN-I response during which ISG transcription can occur without STAT1 and during which STAT1Y701F exerts its inhibitory activity on immunity to *L. monocytogenes* is unclear. However, the biological relevance of the STAT1-independent pathway is documented by antiviral activity of Stat1[−/−] cells against viruses (Dengue virus [19]; MV [18]), by type I IFN-mediated viral pathology in mice [47] and antibacterial macrophage activity against *Le. pneumophila* (Fig 5) [3,30]. Based on our findings and these reports in the literature, we hypothesize that late-phase suppression of the IFN-I response reduces its known detrimental effects on innate resistance to *L. monocytogenes*. However, reduced *L. monocytogenes* growth in Stat1[Y701F] macrophages compared to Stat1[−/−] suggests an additional, cell-autonomous effect of U-STAT1 because cells lacking the type I IFN receptor do not reproduce this phenotype [32] and there is no IFNγ in the experimental system. Extensive microarray analyses are inconsistent with a nuclear activity of the mutant in infected cells, suggesting a hitherto undefined cytoplasmic route through which U-STAT1 influences antibacterial resistance.

In this regard, our observation that *L. monocytogenes* infection causes phosphorylation of the STAT1Y701F mutant at S727 (data not shown) may be of importance, as several reports suggest that at least some biological activities of U-STATs require phosphorylation at this residue. Whereas nuclear S727 phosphorylation by CDK8 enhances transcriptional activity of the tyrosine-phosphorylated STAT1 dimer [48], cytoplasmic S727 phosphorylation of U-STAT1 was linked to diverse biological processes such as the apoptotic response to TNF or the regulation of NK cytotoxicity [49,50]. Therefore, it is tempting to speculate that the cytoplasmic kinase causing S727 phosphorylation of U-STAT1 in *L. monocytogenes*-infected macrophages is part of a STAT1-dependent antibacterial pathway.

In conclusion, our analysis of Stat1[Y701F] cells and mice supports the idea of U-Stat activity and yielded interesting insight into the cross-regulation of STATs. How relevant is this experimental system for Stat signaling in WT cells? Generally speaking, the gain of innate immunity versus STAT1 deficiency seen in our studies is small and hard to extrapolate to a contribution of unphosphorylated STAT1 in WT animals. The best our experimental model can achieve is to reveal a potential of unphosphorylated STAT1, but it cannot provide final proof that this potential is realized in WT cells or animals. The ChIP analysis of Fig 3C demonstrates WT STAT1 at ISREs both early and late after IFNβ treatment and that it co-occupies the same sites with STAT2. This suggests that a STAT1-independent pathway employing STAT2/IRF9 is less prominent in WT cells. Therefore, a selective inhibition of this pathway by unphosphorylated STAT1 can neither be ruled out nor confirmed. From a broader perspective, the finding demonstrating STAT1Y701F mutant inhibition of STAT2 translocation to the nucleus suggests that the subcellular distribution of STAT2 in type I IFN-treated WT cells may be regulated by the relative amount of U-STAT1. Moreover, the scenario studied here applies to patients with Stat1 mutations that inhibit tyrosine phosphorylation [15,16] or to cells in which pathogens disrupt Jak-Stat signal transduction.

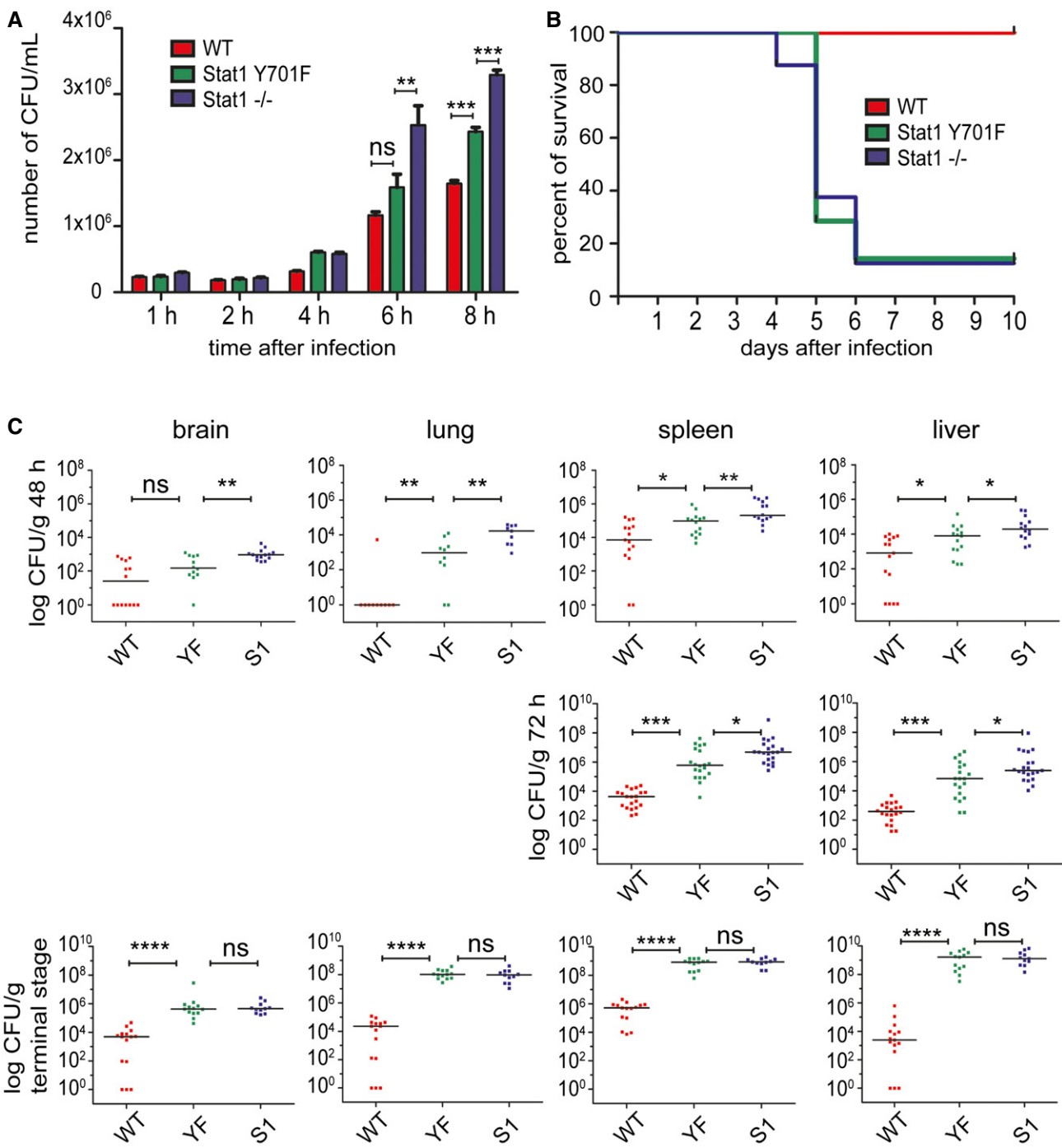

**Figure 7. STAT1Y701F contributes to clearance of *L. monocytogenes* infection.**

A   Impact of Stat1 deficiency or of STAT1Y701F mutation on the growth of *Listeria monocytogenes* in macrophages. Bone marrow-derived macrophages (BMDMs) of wild-type (WT), Stat1$^{Y701F}$, and Stat1$^{-/-}$ mice were infected with *L. monocytogenes* (LO28, MOI 10). Colony-forming unit (CFU) numbers were determined 1, 2, 4, 6, or 8 h after infection by plating on brain–heart infusion (BHI) agar plates. The graph represents biological triplicates and the data are represented as mean values. Error bars represent standard deviation (SD) and asterisks denote statistically significant differences (ns, $P > 0.05$; **$P \leq 0.01$; ***$P \leq 0.001$); $P$-values were calculated using unpaired $t$-test.

B   Survival of mice infected with *L. monocytogenes*. 10 wild-type (WT), Stat1$^{Y701F}$, and Stat1$^{-/-}$ mice per group were infected by intraperitoneal injection of $1 \times 10^2$ viable *L. monocytogenes*. Survival was monitored over 10 days. The study is representative of more than three independent experiments.

C   Organ pathogen burdens of mice infected with *L. monocytogenes*. Wild-type (WT), Stat1$^{Y701F}$ (YF), and Stat1$^{-/-}$ (S1) mice were infected by intraperitoneal injection of $1 \times 10^2$ viable *L. monocytogenes*. Number of colony-forming units (CFU) in organs was determined at 48, 72 h, or at the terminal stage of infection by plating homogenates on brain–heart infusion (BHI) agar plates or Oxford agar plates (for lungs). Dots represent pooled data of three independent experiments. Lines represent the median and asterisks denote statistically significant differences (ns, $P > 0.05$; *$P \leq 0.05$; **$P \leq 0.01$; ***$P \leq 0.001$; ****$P \leq 0.0001$); $P$-values were calculated using Mann–Whitney $U$-test.

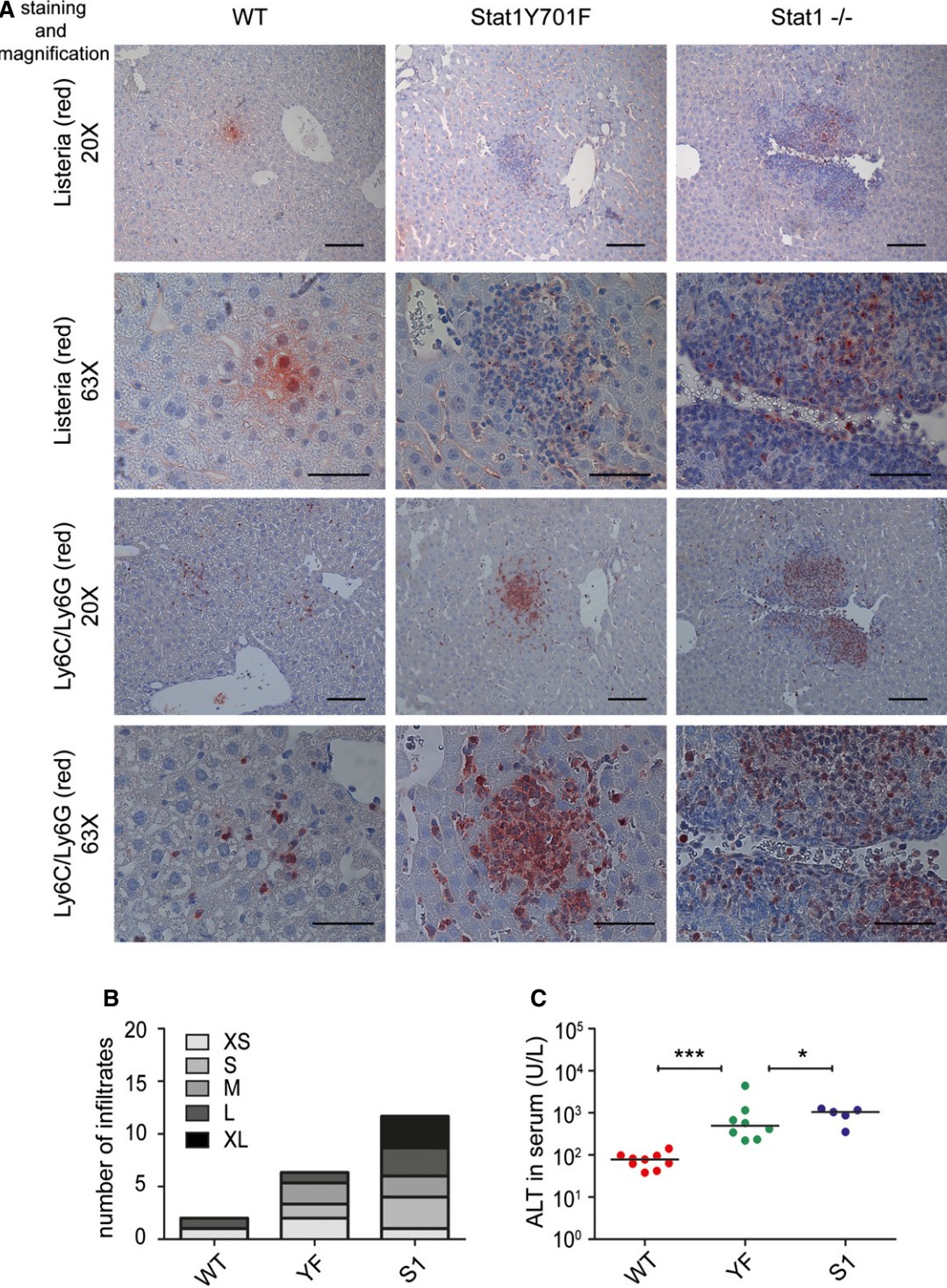

**Figure 8.   Liver inflammation in Stat1$^{Y701F}$ and Stat1$^{-/-}$ mice after infection with *Listeria monocytogenes*.**

A   Immunohistochemical analysis of infection and inflammatory infiltrates. Wild-type (WT), Stat1$^{Y701F}$, and Stat1$^{-/-}$ mice were infected by intraperitoneal injection of 1 × 10$^2$ viable *L. monocytogenes* for 48 h. Liver sections were examined by immunohistochemistry with *L. monocytogenes* or Ly6C/Ly6G-specific antibody. The scale bars on 20× magnification images represent 100 and 50 μm on the 63× magnification images.

B   Quantitative evaluation of inflammatory infiltrates. Infiltrates representing the entire surface of sections from five animals per genotype were counted and categorized according to their size.

C   Liver pathology in infected mice. Wild-type (WT), Stat1$^{Y701F}$ (YF), and Stat1$^{-/-}$ (S1) mice were infected by intraperitoneal injection of 1 × 10$^5$ viable *L. monocytogenes* for 72 h. Serum was collected and tested for ALT activity. Lines represent the median and asterisks denote statistically significant differences (*$P \leq 0.05$; ***$P \leq 0.001$); *P*-values were calculated using unpaired *t*-test.

# Materials and Methods

### Mice

Mice containing an A->T point mutation in exon 23 of the Stat1 gene, causing a change of Y701 to F, were generated following the strategy described for Stat1$^{S727A}$ mice by [51]. In brief, a targeting construct spanning intron 22 through exon 23 was made, inserting a floxed neo cassette in intron 22 and a DT gene after exon 23 for selection of homologous recombinants. The construct was introduced in ES cells and homologous recombinants identified by Southern blot of BglII-restricted genomic DNA. The point mutation was verified by PCR amplification of the tyrosine-containing exon 23 and digestion with restriction enzyme Hinf. The A->T transversion generates an additional Hinf site that divides the PCR amplicon into fragments of 355 and 175 bp length. A positive ES clone (A10) was identified and the neo cassette excised after infection with a Cre-expressing adenovirus. Neo-deleted ES cells were used for blastocyst injection and the generation of chimeric mice. Mice transmitting the mutant allele to their progeny were backcrossed to C57BL/6N mice. A pure C57BL/6N congenic strain was established by marker-assisted selection. C57BL/6N and congenic Stat1$^{-/-}$, Stat2$^{-/-}$, and Irf9$^{-/-}$ mice were housed under SPF conditions [52–54].

All of the animal experiments have been approved by the Vienna University of Veterinary Medicine institutional ethics committee and performed according to protocols approved by the Austrian law (BMWF 68.205/0032-WF/II/3b/2014). General condition and behavior of the animals during the experiments was controlled by FELASA B degree holding personnel. The progress of the disease was monitored every 2–4 h during the "day phase" (7 a.m. to 7 p.m.) or both during the "day" and the "night phase" depending on the condition of the animals. In survival or terminal stage experiments, humane endpoint by cervical dislocation was conducted if death of the animals was expected within next few hours. Animal husbandry and experimentation was performed under the Austrian national law and the ethics committees of the University of Veterinary Medicine Vienna and according to the guidelines of FELASA which match those of ARRIVE.

### Infection of mice and *in vivo* colony-forming unit (CFU) assays

Mice were infected by intraperitoneal injection with the indicated inoculum sizes of the LO28 strain of *L. monocytogenes*. The bacteria were prepared as previously described [55]. For infection, bacteria were washed, diluted in respective concentration in PBS (Sigma), and injected into 8- to 10-week-old, gender-matched C57BL/6N (WT), Stat1$^{Y701F}$, and Stat1$^{-/-}$ mice. The infectious dose was controlled by plating serial dilutions on Oxford agar (Merck Biosciences) or brain–heart infusion (BHI; BD Biosciences) agar plates. For the survival assays, mice were monitored over the course of 10 days. For detection of bacterial loads in liver, spleen, brain, and lungs (CFU assays), mice were sacrificed at the indicated time points, and organs were harvested and homogenized in PBS. The 1:10 serial dilutions were plated on Oxford agar plates (lungs) or BHI agar plates. The colonies were counted after 24-h incubation at 37°C.

### Histology

Mouse organs were harvested, fixed in 4% paraformaldehyde overnight, and dehydrated in 70% ethanol overnight. Samples were further embedded in paraffin and cut on a microtome into 3-μm sections. Immunohistochemical detection of *L. monocytogenes* and of Ly6C/Ly6G$^+$ cells in infected liver tissue was performed as previously described [35]. In brief, sections for the identification of *L. monocytogenes* were incubated in 50% methanol and 3% hydrogen peroxide to inhibit endogenous peroxidase activity and then incubated with pronase (Roche), washed in PBS containing 0.05% Tween (PBS-T), and blocked in 5% normal goat serum. Sections were reacted with primary antibody against *L. monocytogenes* (Abcam) and binding was detected using HRP rabbit/mouse polymer (Dako) and AEC+ high chromogen substrate. The counterstain was done with hematoxylin. For Ly6C/Ly6G (Gr-1) immunohistochemistry, liver sections were incubated in 50% methanol and 3% hydrogen peroxide to inhibit endogenous peroxidase activity and then boiled in 10 mM sodium citrate antigen unmasking solution. The samples were blocked in 3% normal goat serum and stained overnight with primary Ly6C/Ly6G antibody (BD Pharmingen). On the next day, samples were incubated with biotinylated rabbit anti-rat IgG (Vector Laboratories), washed and incubated with ABC reagent (Vector Laboratories). Binding was visualized using AEC+ high sensitivity chromogen substrate (Dako). Samples were counterstained with hematoxylin.

### Analysis of alanine aminotransferase levels (ALT)

Mice were infected with $1 \times 10^5$ LO28 *L. monocytogenes* intraperitoneally and sacrificed 72 h after infection. ALT levels were measured in mice serum using a Roche COBASc11 analyzer (Labor Invitro, Vienna, Austria).

### Cell culture

Bone marrow-derived macrophages (BMDMs) were differentiated from bone marrow isolated from femurs and tibias of 6- to 8-week-old mice. Cells were cultured in DMEM (Sigma-Aldrich) supplemented with 10% of FCS (Sigma-Aldrich), 10% of L929-cell derived CSF-1 and 100 units/ml penicillin, 100 μg/ml streptomycin (Sigma-Aldrich) as previously described [55]. The culture contained >99% of F4/80$^+$ cells.

### *In vitro* colony-forming unit (CFU) assay

*Listeria monocytogenes*: For *in vitro* colony-forming unit (CFU) assays, cells were infected with *L. monocytogenes* LO28 at a multiplicity of infection (MOI) of 10 in antibiotic-free medium. After 1 h, the medium was replaced with medium containing 50 μg/ml of gentamicin in order to eliminate all external bacteria. After one more hour, the medium was exchanged again with medium containing 10 μg/ml gentamicin. At indicated time points, cells were lysed in sterile water and CFUs were determined by plating 1:10 serial dilutions on BHI (BD Biosciences) agar plates. Colonies were counted after 24-h incubation at 37°C.

*Legionella pneumophila*: The *Le. pneumophila* JR32 Fla− (flagellin deficient) strain was grown in AYE (ACES-buffered yeast

extract) broth or on CYE (charcoal yeast extracts) plates as previously described [30]. BMDMs were infected at MOI 0.25 and cells were lysed in sterile water at the indicated time points. Numbers of CFUs were determined by plating 1:10 serial dilutions plated on CYE plates. Colonies were counted after 72 h of incubation at 37°C.

## Preparation of whole-cell lysates and Western blot analysis

$3 \times 10^6$ BMDMs were infected with *L. monocytogenes* at MOI 10 as described above, or treated with 250 IU/ml of IFNβ (PBL interferon source) or 5 ng/ml of IFNγ (Affymetrix, eBioscience). For whole-cell lysates, BMDMs were lysed in 80 μl of Frackelton buffer (10 mM Tris, 30 mM $Na_4P_2O_7$, 50 mM NaCl, 50 mM NaF, 1% Triton X-100, 0.1 mM PMSF, 1 mM DTT, 0.1 mM $Na_3VO_4$, pH 7.5) supplemented with complete protease inhibitors diluted 1:100 (Roche). Proteins were separated on 10% SDS–polyacrylamide gels and blotted onto cellulose membranes (Optitran BA-S 83, GE Healthcare Life Sciences) using a standard semidry protocol (1.5 h, 32 mA per gel). For tissue Western blots, spleens were lysed in 1 ml Frackelton buffer and livers were lysed in five volumes of buffer (50 mM Tris–HCl pH 8, 150 mM NaCl, 1% Triton X-100, 0.1% SDS, 5 mM EDTA, 1 mM EGDA, inhibitors). The following primary antibodies were used: STAT1 C-terminal [56], STAT1 (clone E-23, Santa Cruz), phospho-Y701 STAT1 (Cell Signaling), phospho-Y689 STAT2 (Millipore), STAT2 (Millipore), STAT3 (Cell Signaling), phospho-Y705 STAT3 (Cell Signaling), phospho-Y694 STAT5 (BD Biosciences), STAT5 (Millipore) and tubulin (Sigma). Secondary antibodies were purchased from Li-COR and blots were detected on Odyssey CLx® Infrared Imaging System (Li-COR).

## RNA isolation, cDNA synthesis, and qPCR

$1 \times 10^6$ BMDMs were infected with *L. monocytogenes* at MOI 10 as described above, or treated with 250 IU/ml of IFNβ (PBL interferon source) or 5 ng/ml of IFNγ (Affymetrix, eBioscience). At indicated time points, cells were lysed in RA1 buffer from the NucleoSpin II RNA isolation kit (Macherey-Nagel). Total RNA isolation was further performed according to the manufacturer's instructions. cDNA was synthesized using 200 ng of isolated RNA. qPCR was performed using GoTaq® qPCR Master Mix (Promega) according to the manufacturer's instructions. Samples were amplified on a Mastercycler realplex real-time PCR system (Eppendorf). mRNA levels were calculated and normalized to Gapdh using the $\Delta C_T$ method. Relative fold induction was calculated by normalizing all genotypes and treatments to untreated WT. Sequences of primers are listed in Appendix Table S1.

## Cell transfection

$Stat1^{-/-}$ fibroblasts were grown to 70% confluency in 6-cm dishes. The cells were transfected with 1 μg of each expression plasmid using 8 μl of Turbofect™ reagent (Thermo Scientific). Twenty-four hours later, the cells were treated with IFNβ for 48 h, followed by extraction of RNA as described above.

## Immunofluorescence

$2 \times 10^5$ BMDMs were seeded on glass cover slides, treated with 250 IU/ml IFNβ, and fixed with 3% paraformaldehyde for 20 min at room temperature. Cells were permeabilized with 0.1% saponin in 0.5 M NaCl PBS. Blocking and incubation with STAT2 primary antibody [3] and secondary anti-rabbit Alexa Fluor® 488 (Life Technologies) were done in 0.1% saponin and 1% BSA in 0.5 M NaCl PBS. Samples were mounted in DAPI-containing mounting media Dapi-Fluoromount-G (SouthernBiotech). Confocal images were acquired using a Zeiss LSM 710 microscope with 63X (NA 1.4) oil objectives. Images were processed and analyzed using the ImageJ software and relative fluorescence was calculated according to the corrected total cell fluorescence (CTCF) formula (CTCF = Integrated density − (area of selected cell × mean fluorescence background readings)) as previously described [57,58].

## ChIP-seq, ChIP, and ChIP-reChIP

Chromatin immunoprecipitation (ChIP) using DynaBeads Protein G (Invitrogen) was performed as described [59]. BMDMs were treated with 250 IU/ml of IFNβ. ChIP was performed using STAT1 (clone E-23, Santa Cruz) or STAT2 antibody (clone C-20, Santa Cruz). Levels of precipitated chromatin were measured by qPCR as described above. Primers used for qPCR are listed in Appendix Table S1. Data were normalized to input and to the sample from untreated wild-type cells. For ChIP-seq, precipitated chromatin was sonicated into 200- to 300-bp-long fragments. 5–10 ng of DNA precipitated by ChIP was used as the starting material for the generation of sequencing libraries using the KAPA library preparation kit for Illumina systems. Completed libraries were quantified with a Bioanalyzer dsDNA 1000 assay kit (Agilent) and a qPCR NGS library quantification kit (KAPA). Cluster generation and paired-end sequencing was achieved with a HiSeq 2000 system with a read length of 100 bp according to the manufacturer's guidelines (Illumina). Samples were multiplexed using unique adaptors; all the untreated samples were run on the first lane and the treated ones on the second lane of the flow cell. ChIP-reChIP experiments were preformed as previously described [60] using STAT1 (clone E-23, Santa Cruz) or STAT2 antibody (clone C-20, Santa Cruz). In short, the immune complexes were eluted with 10 mM DTT for 40 min at room temperature with agitation after which they were diluted 40-fold in ChIP dilution buffer and re-immunoprecipitated.

## ChIP-Seq analysis

Quality-based trimming was performed at the 3′ end of raw reads using the "trim-fastq.pl" script of the PoPoolation toolbox [61], where all trimmed reads with length less than 30 bp were discarded. Quality controlled reads were mapped to the mouse genome (UCSC, mm10) using Bowtie [62] where all non-unique alignments were discarded. Post-alignment filtering: Reads mapped in a proper pair were selected and PCR duplicates were removed using SAMtools [63]. Peak calling was performed using MACS [64] with default parameters. Significant peaks were annotated to the nearest genes using PeakAnalyzer [65]. ChIP-Seq data for Stat1$^{Y701F}$ mutant and Stat1$^{-/-}$ samples are deposited at ArrayExpress under accession number E-MTAB-3597 (https://www.ebi.ac.uk/arrayexpress/experiments/E-MTAB-3597/) and for wild-type Stat1 sample are deposited at ArrayExpress under accession number E-MTAB-2972 (https://www.ebi.ac.uk/arrayexpress/experiments/E-MTAB-2972/).

## Microarray

Total RNA from mouse BMDMs treated with *L. monocytogenes* for 6, 12, or 24 h was purified using TRIzol (Invitrogen, USA) and RNeasy Mini Kit (Qiagen, Germany). The RNA samples were collected from 3 sets of independent experiments. Samples were prepared by the Genomics Core of Lerner Research Institute, Cleveland Clinic, using 1 g of total RNA and Illumina Mouse Ref-8 v2 Expression BeadChips (Illumina Inc. USA). After filtering to remove probes expressed at or below background levels in the WT, 7644 probes remained. The log expression level of each remaining probe was analyzed with a linear model with genotype, time point, and replicate as factors. The mean expression of the $Stat1^{Y701F}$ genotype was contrasted with that of the WT and $Stat1^{-/-}$. Subsequently a Benjamini–Hochberg false discovery rate analysis was performed independently for the three comparisons among genotypes. Microarray data are deposited at ArrayExpress under accession number E-MTAB-3598 (https://www.ebi.ac.uk/arrayexpress/experiments/E-MTAB-3598/).

## Statistical analysis

Bacterial loads of organs and ALT levels in serum were compared using the Mann–Whitney $U$-test and middle values represent medians. Bacterial loads in *in vitro* CFU assays were compared with the Student's $t$-test and bars on the graph represent the mean values with error bars that represent standard deviation (SD). mRNA expression data also represent the mean values with standard error of the mean (SEM). The differences in mRNA expression data were compared using the ratio $t$-test. All statistical analyses were performed using the GraphPad Prism (GraphPad) software. Asterisks denote statistical significance as follows: ns, $P > 0.05$; *$P \leq 0.05$; **$P \leq 0.01$; ***$P \leq 0.001$.

**Expanded View** for this article is available online.

## Acknowledgements

We thank Ines Jeric for valuable suggestions regarding experimental procedures. We gratefully acknowledge Orest Kuzyk for help with the quantification of immunofluorescence images. Funding was provided by the Austrian Science Fund (FWF) through grant SFB-28 (to V.S., M.M., and T.D.). A.M. was supported by the FWF through the doctoral program Molecular Mechanisms of Cell Signaling. Next-generation sequencing was performed at the Vienna Biocenter's CSF NGS Unit (http://www.csf.ac.at/home/).

## Author contributions

AM performed the experiments; EP and DS participated in revision of the manuscript; HC, CV and PS analyzed the ChIP-seq and microarray data; GRS, RS and TD generated the mouse models; MM, VS, CS and TD supervised the experiments and gave valuable input; AM, CS and TD wrote the manuscript.

## Conflict of interest

The authors declare that they have no conflict of interest.

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
