## [Review Process File · EMBO Reports]

Manuscript EMBO-2015-40726

Response to interferons and antibacterial innate immunity in the absence of tyrosine-phosphorylated STAT1

Andrea Majoros, HyeonJoo Cheon, Claus Vogl, Priyank Shukla, George Stark, Veronika Sexl,
Robert Schreiber, Christian Schindler, Mathias Müller, Thomas Decker

Corresponding author: Thomas Decker, University of Vienna

Review timeline:

Submission date:	22 May 2015
Editorial Decision:	06 July 2015
Revision received:	03 December 2015
Editorial Decision:	23 December 2015
Revision received:	04 January 2016
Accepted:	13 January 2016

Transaction Report:

1st Editorial Decision

06 July 2015

Thank you for your submission to EMBO reports. We have now received reports from two of the three referees that were asked to evaluate your study, which can be found at the end of this email. As they are in fair agreement, I am making a decision now to avoid further delaying the process. As you will see, both referees find the topic of interest and in principle suitable for us, although they raise a few issues that should be addressed during revision of your study.

All referees concerns are pertinent and should be addressed for a successful revision. Please note that it is EMBO reports policy to undergo one round of revision only and thus, acceptance of your study will depend on the outcome of the next, final round of peer-review.

Revised manuscripts must be submitted within three months of a request for revision unless previously discussed with the editor; they will otherwise be treated as new submissions.

- a complete author checklist, which you can download from our author guidelines (<http://embor.embopress.org/authorguide#revision>)
- a letter detailing your responses to the referee comments in Word format (.doc)
- a Microsoft Word file (.doc) of the revised manuscript text
- editable TIFF or EPS-formatted figure files in high resolution
- a separate PDF file of any Supplementary information (in its final format)

EMBO reports now accommodates the inclusion of extra figures (up to five) in the online version of the manuscript. These are presented in an expandable format inline in the main text so that readers who are interested can access them directly as they read the article. They are also provided for download in a separate typeset PDF to accompany the Article PDF. These should be those of particular value to specialist readers, but which are not required to follow the main thread of the paper (and not additional controls or reagent optimization). These should be labeled expanded view, and the rest supplementary.

We also encourage the publication of original source data -particularly for electrophoretic gels and blots, but also for graphs- with the aim of making primary data more accessible and transparent to the reader. If you agree, you would need to provide one PDF file per figure that contains the original, uncropped and unprocessed scans of all or key gels used in the figures and an Excel sheet or similar with the data behind the graphs. The files should be labeled with the appropriate figure/panel number, and the gels should have molecular weight markers; further annotation could be useful but is not essential. The source files will be published online with the article as supplementary "Source Data" files and should be uploaded when you submit your final version. If you have any questions regarding this please contact me.

I look forward to seeing a revised form of your manuscript when it is ready. In the meantime, please contact me if I can be of any assistance.

REFEREE REPORTS

Referee #1:

Majoros et al have addressed the question of the role of noncanonical, i.e. tyrosine phosphorylation independent, STAT1 activity in innate immunity by generating a Stat1YF mutant knock-in mouse. Homozygous Stat1YF mice showed significantly reduced Stat1 expression and absence of immediate type I and II interferon transcriptional responses. Surprisingly, the delayed expression of IFN-I-stimulated genes, mediated by STAT2-IRF9 complex, was inhibited. Stat1YF cells were more susceptible to *Legionella pneumophila* infection but less susceptible to *Listeria monocytogenes* infection than Stat1 knockout cells. The results support a subtle role for non-tyrosine phosphorylated Stat1 in tonic regulation of Stat1 protein levels and in innate defence against bacteria.

The study is a carefully performed and well-written and I have the following comments.

The most significant effect of Stat1YF mutation is observed on Stat1 protein levels which highlights the importance of phosphotyrosine-dependent autoregulation of the gene. The other side of the coin is that the differences in Stat1 protein levels make the interpretation of the mechanistic and functional studies is more difficult.

The results in Fig 3 lead to the conclusion that the unexpected inhibition of STAT2-IRF9 complex by Stat1YF is due to cytoplasmic interaction and prevention of nuclear localization of STAT2. This could be quite easily tested with the constructs in hand.

Fig 5, the effect of IFN-I on growth of *Listeria* in macrophages looks quite small. Can it be concluded with three biological replicates that there is a difference between Stat1^{-/-} and Stat1YF in Fig 5B?

In Fig 6 it would be good to show Stat1 levels

As the authors note, the small Stat1YF protein amount pose a challenge, particularly for the in vivo studies but a small effect compared to knockout on bacterial load in different organs is observed. However, in Fig 7B the lack of effect on survival between knockout and Stat1YF (despite presence of Stat1YF protein vs lack of protein in KO) raises a question how important the function of Stat1YF is in innate immunity. I think the text should more clearly reflect this aspect, particularly in abstract.

Referee #2:

Decker and colleagues describe the phenotype of mice expressing a Stat1 mutant that cannot be tyrosine701-phosphorylated, aka inactive Stat1 (Stat1YF). Inactive Stat1 is the precursor of the activated variant that has numerous roles in innate immunity and interferon signalling. Whether inactive STAT1 has additional independent roles is essentially unknown. Their generally well-executed study addresses this important question and provides some interesting insights into the transcriptional regulation of STAT1/2 genes and the role of inactive STAT1 in immunity. The main finding of this work is that mice that express inactive Stat1 differ from those that lack Stat1 altogether. The differences are subtle yet in part significant. The authors conclude that inactive Stat1 inhibits the signalling via the Stat2/Irf9 complex which is formed late in the type I interferon response and generally augments it. This adds to the growing body of evidence indicating that inactive Stats contribute to and/or variegate cytokine signalling. To my knowledge this is the first study that takes the analysis of inactive Stats to the level of transgenic animals. This is, therefore, relevant work of which I think three aspects need attention.

One is the experimental system, namely the choice of Stat1YF as a model for inactive Stat1. It is important to remember that Stat1YF is not simply an inactive Stat1 and precursor of active wild type Stat1 but a Stat1 variant that cannot be activated. This therefore is an artificial situation and unlike wild type Stat1 and findings with Stat1YF do not necessarily reproduce the activities simply of inactive wild type Stat1. Thus, caution is advised when making such extrapolations (e.g. lines 346-348). As the authors rightly point out Stat1YF resembles the situation in patients with mutated Stat1, and can help to better understand their condition, despite some differences to the reported human disease. The distinction between inactive and inactivatable is further relevant when it comes to in vivo analyses and the differences to Stat1 deficient animals. An example is the interpretation of *Listeria* infections, where type I interferon has disease promoting effects. In comparison to Stat1 deficient cells the inactive Stat1 reduces bacterial replication in macrophages in vitro, and the bacterial loads in infected Stat1YF animals likewise are lower in comparison to Stat1 deficient mice. However, the differences disappear during the course of infection and mortality is the same in the two mouse strains. The authors explain this as being the result of the strongly reduced interferon-beta production in both Stat1 deficient and Stat1YF mice and cells. This could be tested in vitro by adding interferon but this has not been done. An alternative explanation could be the known compensatory signalling in Stat1 deficient mice that is likely to be suppressed in the Stat1YF animals for example in response to type II interferon. As long as only in vitro cell assays and stimulation with type I interferon are considered, this may be of little relevance indeed. However, this could be different in the whole animal, where the results cannot be interpreted solely in light of the effects of Stat1YF on type I interferon. The authors touch briefly upon compensatory signalling, however only in the context of type I but not type II interferon. A model of inactive Stat1 that does not suppress potential compensatory signalling could have been chosen (SH2 mutant) but it is too late for this now. I would encourage the authors to address this potential limitation of their model in the discussion at least, which at present is centred almost exclusively on type I interferon.

Another concern is with Figure 4. The results in 4A do not show clearly enough the stated differences in Stat2 nuclear translocation. The quantification in 4B moreover is not well described. It is not clear which units are used, and whether the numbers given for the nuclear Stat2 are absolute or relative values, or whether any normalization was applied. Since there are considerable differences in Stat2 expression between the different cells these considerations require clarification.

Finally the data in Figure 6B need attention. Here the authors address the role of serine phosphorylation for the functions of inactive Stat1. Serine phosphorylation can enhance Stat1 activity; it has been reported to occur via at least two separate mechanisms, either via CDK8 in response to interferons and via MAP kinases during microbial infection. It is of particular relevance for this work that these mechanisms have been shown to take place in different cellular compartments, namely in the nucleus (CDK8) or in the cytoplasm (MAPK). Since it is demonstrated here that Stat1YF does not enter the nucleus and the authors conclude that this is how it can inhibit interferon signalling, it would be expected that Stat1YF is serine phosphorylated in microbe-infected cells, but not in interferon-stimulated cells. The authors accordingly state in the text (line 223) that the result of Figure 6B showing serine phosphorylation of Stat1YF "is in line with experimental evidence that IFN-dependent STAT1 serine-phosphorylation is catalysed by chromatin associated CDK8, whereas during infection the p38 MAPK pathway establishes an additional, cytoplasmic route" for Stat1 serine phosphorylation. However, Figure 6B shows only the results for *Listeria*-

infected cells, but not for interferon treatment, such that nothing can be said about the role of CDK8 in Stat1YF phosphorylation. Given the importance of serine phosphorylation for understanding the functioning of Stat1 in general and inactive Stat1 in particular, this question that can be addressed with relatively little experimental effort should be resolved.

1st Revision - authors' response

03 December 2015

We appreciate the constructive comments made by the referees to improve our manuscript. Our replies and the changes in the revised manuscript, including new data, are listed in our point-by-point reply.

Referee #1:

The study is a carefully performed and well-written and I have the following comments.

1. The most significant effect of Stat1YF mutation is observed on Stat1 protein levels which highlights the importance of phosphotyrosine-dependent autoregulation of the gene. The other side of the coin is that the differences in Stat1 protein levels make the interpretation of the mechanistic and functional studies is more difficult.

We agree with the referee and explain the problem in our manuscript: 'Due to the strong decrease in STAT1 amounts caused by Y701F mutation with respect to WT, the most important read-out of these experiments is a potential gain of function in comparison to Stat1^{-/-} mice.' (new manuscript, p. 8 line 231-233).

We realize that due to reduced STAT1Y701F amounts loss of function against cells expressing STAT1wt is not interpretable. Accordingly, biologically meaningful data are based on gain-of-function versus Stat1^{-/-} cells and animals.

2. The results in Fig 3 lead to the conclusion that the unexpected inhibition of STAT2-IRF9 complex by Stat1YF is due to cytoplasmic interaction and prevention of nuclear localization of STAT2. This could be quite easily tested with the constructs in hand.

We performed the experiment as suggested. Indeed, transfection of STAT1Y701F together with STAT2 into STAT1^{-/-} fibroblasts results in suppression of gene expression compared to the level achieved by STAT2 alone, or by STAT2 together with STAT1 WT (new figure 3D, text on p. 6 line 182-187 and lines 466-469).

3. Fig 5, the effect of IFN-I on growth of Listeria in macrophages looks quite small. Can it be concluded with three biological replicates that there is a difference between Stat1^{-/-} and Stat1YF in Fig 5B?

The effect on *Legionella* replication is not large, but represents a significant rescue of what can be achieved by IFN β treatment in wt cells. In order to improve the credibility of this data, we have performed 3 more replicates of the experiment. The biological outcome remains the same, but the level of statistical significance is improved (new figure 5 and legend to new figure 5).

4. In Fig 6 it would be good to show Stat1 levels.

The revised figure 6 now includes this control, as well as Stat1^{-/-} cells as negative control. Please note our reply to point 5 made by referee 2. For reasons outlined in our reply and the data shown in figures R2 and R3 we omitted the data showing STAT1 phosphorylation at S727.

5. As the authors note, the small Stat1YF protein amount pose a challenge, particularly for the in vivo studies but a small effect compared to knockout on bacterial load in different organs is observed. However, in Fig 7B the lack of effect on survival between knockout and Stat1YF (despite presence of Stat1YF protein vs lack of protein in KO) raises a question how important the function

of Stat1YF is in innate immunity. I think the text should more clearly reflect this aspect, particularly in abstract.

The abstract now contains the statement that ‘...*Listeria monocytogenes* grew less robustly in Stat1^{Y701F} macrophages and mice compared to Stat1^{-/-} counterparts, but STAT1^{Y701F} did not rescue the animals’. Furthermore we state a ‘potential’ of STAT1Y701F to contribute to antibacterial immunity (p. 2 line 43/44). Since the abstract word limit did not allow further clarification we state in the concluding (last) paragraph of the discussion that ‘...the gain of innate immunity versus STAT1 deficiency seen in our studies is small and hard to extrapolate to a contribution of unphosphorylated STAT1 in WT animals’. (p. 11 line 341-345).

Referee #2:

This is, therefore, relevant work of which I think three aspects need attention.

1. One is the experimental system, namely the choice of Stat1YF as a model for inactive Stat1. It is important to remember that Stat1YF is not simply an inactive Stat1 and precursor of active wild type Stat1 but a Stat1 variant that cannot be activated. This therefore is an artificial situation and unlike wild type Stat1 and findings with Stat1YF do not necessarily reproduce the activities simply of inactive wild type Stat1. Thus, caution is advised when making such extrapolations (e.g. lines 346-348).

We agree with the referee. In fact the statement ends with the sentence that based on our data ‘...a selective inhibition of this pathway (*i.e. late, STAT2/IRF9-dependent*) by unphosphorylated STAT1, can neither be ruled out nor confirmed.’ (p. 11 line 348). For further clarity we added a sentence earlier in the same paragraph stating that ‘...the gain of innate immunity versus STAT1 deficiency seen in our studies is small and hard to extrapolate to a contribution of unphosphorylated STAT1 in WT animals’ (lines 341-345).

2. In comparison to Stat1 deficient cells the inactive Stat1 reduces bacterial replication in macrophages in vitro, and the bacterial loads in infected Stat1YF animals likewise are lower in comparison to Stat1 deficient mice. However, the differences disappear during the course of infection and mortality is the same in the two mouse strains. The authors explain this as being the result of the strongly reduced interferon-beta production in both Stat1 deficient and Stat1YF mice and cells. This could be tested in vitro by adding interferon but this has not been done.

We did not mean to say that the strongly reduced IFN β synthesis explains the effects of STAT1Y701F. Rather, we tried to state that any potential effects IFN β might have in STAT1Y701F cells will be mitigated by the fact that IFN β synthesis is strongly decreased. The suggested experiment to treat *Listeria*-infected STAT1^{Y701F} cells with IFN β has been tried without producing any evidence for an altered course of infection. We tried to make our reasoning clearer by improving the text: ‘...the strongly reduced production of type I IFN is likely to obscure potential effects IFN-I might exert on the innate response of Stat1^{Y701F} cells and mice to *L. monocytogenes*. It may also preclude significant alterations of gene expression when compared to STAT1-deficiency.’ (p. 7, lines 221-224).

3. An alternative explanation could be the known compensatory signalling in Stat1 deficient mice that is likely to be suppressed in the Stat1YF animals for example in response to type II interferon. As long as only in vitro cell assays and stimulation with type I interferon are considered, this may be of little relevance indeed. However, this could be different in the whole animal, where the results cannot be interpreted solely in light of the effects of Stat1YF on type I interferon. The authors touch briefly upon compensatory signalling, however only in the context of type I but not type II interferon. A model of inactive Stat1 that does not suppress potential compensatory signalling could have been chosen (SH2 mutant) but it is too late for this now. I would encourage the authors to address this potential limitation of their model in the discussion at least, which at present is centered almost exclusively on type I interferon.

Although activation of STATs 3-6 was not directly investigated, figure 2 suggests that compensation by any of these does not lead to a rescue of gene expression in response to IFN γ (figure 2). That said, we agree with the referee that this or other compensation mechanisms might occur in mice. We also agree that mice expressing a STAT1 SH2 domain mutant would be interesting to study. When deciding for STAT1^{Y701F} we reasoned that an SH2 domain mutant would not be able to interact with IFN receptor complexes to be phosphorylated at Y701F and would essentially phenocopy the Y701F mutation (see also Kovarik et al., EMBO J. (2001) 20 : 91-100).

To clarify and emphasize the difference between inactive (i.e. unphosphorylated WT) and STAT1Y701F we extended the final paragraph of the discussion as follows: 'Generally speaking, the gain of innate immunity versus STAT1 deficiency seen in our studies is small and hard to extrapolate to a contribution of unphosphorylated STAT1 in wt animals. The best our experimental model can achieve is to reveal a potential of unphosphorylated STAT1, but it cannot provide final proof that this potential is realized in WT cells or animals.' (p. 11 lines 341-345.)

4. Another concern is with Figure 4. The results in 4A do not show clearly enough the stated differences in Stat2 nuclear translocation. The quantification in 4B moreover is not well described. It is not clear which units are used, and whether the numbers given for the nuclear Stat2 are absolute or relative values, or whether any normalization was applied. Since there are considerable differences in Stat2 expression between the different cells these considerations require clarification.

The revised manuscript emphasizes the evidence for our interpretation. First, we have made an effort to describe the quantification method better (p. 15 lines 478-480) and revised the graph in figure 4B accordingly (labeling of the y-axis). Second, the quantification shown in figure 4B now includes an additional experiment, which caused no substantial change (mentioned in the figure legend). Third, we performed an experiment suggested by referee 1 showing that the inhibition of the late, STAT2/IRF9-dependent pathway can be reproduced by transfecting STAT1Y701F into STAT1-deficient fibroblasts (new figure 3D, text on p. 6 lines 182-187). We also provide the referee with immunofluorescence images of further cells stained as in figure 4A to demonstrate reproducibility and robustness of our findings (Fig. R1A and R1B).

5. Finally the data in Figure 6B need attention. Here the authors address the role of serine phosphorylation for the functions of inactive Stat1. Serine phosphorylation can enhance Stat1 activity; it has been reported to occur via at least two separate mechanisms, either via CDK8 in response to interferons and via MAP kinases during microbial infection. It is of particular relevance for this work that these mechanisms have been shown to take place in different cellular compartments, namely in the nucleus (CDK8) or in the cytoplasm (MAPK). Since it is demonstrated here that Stat1YF does not enter the nucleus and the authors conclude that this is how it can inhibit interferon signaling, it would be expected that Stat1YF is serine phosphorylated in microbe-infected cells, but not in interferon-stimulated cells. The authors accordingly state in the text (line 223) that the result of Figure 6B showing serine phosphorylation of Stat1YF "is in line with experimental evidence that IFN-dependent STAT1 serine-phosphorylation is catalysed by chromatin associated CDK8, whereas during infection the p38 MAPK pathway establishes an additional, cytoplasmic route" for Stat1 serine phosphorylation. However, Figure 6B shows only the results for Listeria-infected cells, but not for interferon treatment, such that nothing can be said about the role of CDK8 in Stat1YF phosphorylation. Given the importance of serine phosphorylation for understanding the functioning of Stat1 in general and inactive Stat1 in particular, this question that can be addressed with relatively little experimental effort should be resolved.

The requested experiment has given us some headache. With a new supplier (Li-Cor) of the secondary antibody we detect a low level of S727 phosphorylation of the STAT1Y701F mutant after both IFN β and IFN γ treatment (upper panels of Fig. R2, a more sensitive membrane increases the effect). This contrasted with earlier results and obliged us to perform direct comparisons with the antibodies used in our initial experiments (Rockland). We observe a reproducible difference between the two reagents (lower panel of Fig. R2).

Thus, under appropriate experimental conditions we detect some S727 phosphorylation by IFN, raising the question whether this is a nuclear event, or cytoplasmic and due to low-level p38MAPK activation as proposed by Brian Williams' lab (Goh, K.C., Haque, S.J. & Williams, B.R.G., 1999).

p38 MAP kinase is required for STAT1 serine phosphorylation and transcriptional activation induced by interferons. *Embo J*, 18, pp.5601–5608.). To address this point we added inhibitors of p38MAPK (SB 203580) or CDK8 (Flavopiridol). The results shown in figure R3 suggest that SB203580 indeed suppresses IFN-mediated S727 phosphorylation, albeit incompletely, whereas Flavopiridol shows no detectable effect.

By and large, these results confirm our notion that there is no nuclear, CDK8-dependent S727 phosphorylation. However, this is now a very complicated experiment to emphasize a point that is more convincingly made by the ChIP-Seq analysis shown in figure S3. It also suffers from the fact that amounts of S727 phosphorylated STAT1Y701F are very low and the inhibitory effects of SB203580 correspondingly small. For these reasons we hope the referee agrees to our decision to omit this experiment from the manuscript. We feel it does not contribute enough new information to be of essential character.

Figure R1: Analysis of STAT2 nuclear translocation by immunofluorescence – additional images. Bone marrow derived macrophages (BMDMs) of wild type (WT), Stat1^{Y701F} (YF), Stat1^{-/-} (S1), Stat2^{-/-} (S2) and IRF9^{-/-} (IRF9) mice were seeded on cover slips and stimulated with 250 IU/mL of IFNβ for 30 min or 24 h. The cells were fixed and stained for STAT2 specific antibody followed by Alexafluor® 488 conjugated secondary antibody (green). Nuclei were stained with DAPI (blue).

Figure R2: Western blot analysis of STAT1 S727 phosphorylation. Bone marrow derived macrophages (BMDMs) of wild type (WT), Stat1^{Y701F} (YF) or Stat1^{-/-} (S1) mice were treated with 250 IU/mL of IFN β or 5 ng/mL of IFN γ for 0.5 or 2 h. Whole cell extracts were collected and tested in western blot for levels of phosphorylation of STAT1 (pS727) and total STAT1 levels. The blots are representative of more than 3 independent experiments.

Figure R3: Stat1S727 phosphorylation in Stat1Y701F mutant is inhibited by p38 MAPK inhibitor (SB203580 - SB) but not by CDK inhibitor (flavopiridol - FP). Bone marrow derived macrophages (BMDMs) of wild type (WT), Stat1^{Y701F} (YF) or Stat1^{-/-} (S1) mice were treated with SB (20 μ M) or FP (500 nM) inhibitors. One hour later the cells were stimulated with 250 IU/mL of IFN β or 5 ng/mL of IFN γ for 0.5 h. Whole cell extracts were collected and tested in western blot for levels of phosphorylation of STAT1 (pS727) and total STAT1 levels. The blots are representative of more than 3 independent experiments.

Thank you for the submission of your revised manuscript to EMBO reports. We have now received the full set of referee reports that is copied below.

As you will see, both referees are positive about the study and referee 2 requests only minor changes, i.e., to include a statement on the role of serine 727 phosphorylation in the discussion.

From the editorial side, there are also a few things that we need before we can proceed with the official acceptance of your study.

- You have currently submitted your study as Scientific Report. This format should not exceed 25,000 characters and 5 main figures. As your manuscript has 8 figures I suggest submitting the revised version as Article. In this case the Results and Discussion section can stay as it is now. For further information see also our guide for authors.

- Supplementary data: You can submit up to 5 images as Expanded View, if you wish. Please follow the nomenclature Figure EV1, Figure EV2 etc. The figure legend for these should be included in the main manuscript document file in a section called Expanded View Figure Legends after the main Figure Legends section.

If the additional Supplementary material is supplied as Appendix, please provide a single pdf file labeled Appendix. The Appendix includes a table of content on the first page, all figures and their legends. Please follow the nomenclature Appendix Figure Sx throughout the text and also label the figures according to this nomenclature. For more details please refer to our guide to authors.

- Regarding data quantification, can you please specify the test used to calculate p-values in the respective figure legends? This information is currently incomplete and must be provided in the figure legends. Please also include scale bars in all microscopy images.

- As a standard procedure, we edit the title and abstract of manuscripts to make them more accessible to a general readership. Please find the edited versions below my signature and let me know if you do NOT agree with any of the changes.

Abstract:

"Signal transducer and activator of transcription 1 (STAT1) plays a pivotal role in the innate immune system by directing the transcriptional response to interferons (IFN). Stat1 is activated by Janus kinase (JAK)-mediated phosphorylation of Y701. To determine whether STAT1 contributes to cellular responses without this phosphorylation event, we generated mice with Y701 mutated to a phenylalanine (Stat1Y701F). We show that heterozygous mice do not exhibit a dominant negative phenotype. Homozygous Stat1Y701F mice show a profound reduction in Stat1 expression, highlighting an important role for basal IFN-dependent signaling. The rapid transcriptional response to type I IFN (IFN-I) and type II IFN (IFN γ) is absent in Stat1Y701F cells. Intriguingly, STAT1Y701F suppresses the delayed expression of IFN-I-stimulated genes (ISG) observed in Stat1^{-/-} cells, mediated by the STAT2- IRF9 complex. Thus, Stat1Y701F macrophages are more susceptible to *Legionella pneumophila* infection than Stat1^{-/-} macrophages. *Listeria monocytogenes* grow less robustly in Stat1Y701F macrophages and mice compared to Stat1^{-/-} counterparts, but STAT1Y701F is not sufficient to rescue the animals. Our studies are consistent with a potential contribution of Y701-unphosphorylated STAT1 to innate antibacterial immunity."

REFEREE REPORTS

Referee #1:

The revised manuscript provides adequate answers to my questions.

Referee #2:

The authors have addressed the queries and provided convincing answers. I would like them to reconsider, however, the decision to entirely purge from the manuscript their extensive yet ultimately unsuccessful attempts to clarify the role of serine 727. This would not require a figure, but a sentence or two in the discussion in my view would be appropriate and warranted, not least in light of the space and attention the authors had previously devoted to this aspect. In connection with this, the manuscript still refers to a Figure 6c that no longer exists.

2nd Revision - authors' response

04 January 2016

Reply to referee 2:

As requested by the referee we discuss our results regarding STAT1 serine phosphorylation (p. 11, lines 339-347). We also corrected the reference to figure 6C (p. 11 line 322).

Changes required by the editor:

1. We submitted the revised manuscript as an article.
2. We chose the option of displaying our supplemental figures as extended view and made the corresponding changes in the manuscript (figure labeling, expanded view figure legends after the main figure legends, in-text references). The only appendix remaining is 'appendix table1', which lists primer sequences.
3. The calculation of p values is explained in materials and methods (p. 18 lines 534-541) and has now also been inserted into figure legends where appropriate. Scale bars are included into microscopy images and their scale indicated in the figure legends.
4. We agree with changes in the abstract.
5. A synopsis and synopsis image have been included.

3rd Editorial Decision

13 January 2016

I am very pleased to accept your manuscript for publication in the next available issue of EMBO reports. Thank you for your contribution to our journal.